# Escaping Saddle Points in Nonconvex Minimax Optimization via Cubic-Regularized Gradient Descent-Ascent

## Abstract

The gradient descent-ascent (GDA) algorithm has been widely applied to solve nonconvex minimax optimization problems. However, the existing GDA-type algorithms can only find first-order stationary points of the envelope function of nonconvex minimax optimization problems, which does not rule out the possibility to get stuck at suboptimal saddle points. In this paper, we develop Cubic-GDA – the first Newton-type GDA algorithm for escaping strict saddle points in nonconvex-strongly-concave minimax optimization. Specifically, the algorithm uses gradient ascent to estimate the second-order information of the minimax objective function, and it leverages the cubic regularization technique to efficiently escape the strict saddle points. Under standard smoothness assumptions on the objective function, we show that Cubic-GDA admits an intrinsic potential function whose value monotonically decreases in the minimax optimization process. Such a property leads to a desired global convergence of Cubic-GDA to a second-order stationary point at a sublinear rate. Moreover, we analyze the convergence rate of Cubic-GDA in the full spectrum of a gradient dominant-type nonconvex geometry. Our result shows that Cubic-GDA achieves an orderwise faster convergence rate than the standard GDA for a wide spectrum of gradient dominant geometry. Our study bridges minimax optimization with second-order optimization and may inspire new developments along this direction.

## 1 Introduction

Nonconvex minimax optimization is a popular optimization framework that has broad applications in modern machine learning, including game theory (Ferreira et al., 2012), generative adversarial networks (Goodfellow et al., 2014), adversarial training (Sinha et al., 2017), reinforcement learning (Qiu et al., 2020; Ho and Ermon, 2016; Song et al., 2018), etc. A standard nonconvex minimax optimization problem is shown below, where $f$ is a smooth nonconvex function in $x$.

$$\min_{x \in \mathbb{R}^m} \max_{y \in \mathbb{R}^n} f(x, y). \tag{P}$$

In the existing literature, many optimization algorithms have been developed to solve different types of minimax problems. Among them, a simple and popular algorithm is the gradient descent-ascent (GDA), which alternates between a gradient descent update on $x$ and a gradient ascent update on $y$ in each iteration. Specifically, the global convergence of GDA has been established for minimax problems under various types of global geometries, such as convex-concave-type geometry ($f$ is convex in $x$ and concave in $y$) (Nedić and Ozdaglar, 2009; Du and Hu, 2019; Mokhtari et al., 2020; Zhang and Wang, 2021), bi-linear geometry (Neumann, 1928; Robinson, 1951) and Polyak-Łojasiewicz geometry (Nouiehed et al., 2019; Yang et al., 2020), yet these geometries are not satisfied by general nonconvex minimax problems. Recently, many studies proved the convergence of GDA in nonconvex minimax optimization for both nonconvex-concave problems (Lin et al., 2020; Nouiehed et al., 2019; Xu et al., 2020d) and nonconvex-strongly-concave problems (Lin et al., 2020; Xu et al., 2020d; Chen et al., 2021). In these studies, it has been shown that GDA converges sublinearly to a stationary point where the gradient of a certain envelope function of the minimax problem vanishes.

Although GDA can find first-order stationary points of nonconvex minimax problems, such a type of convergence guarantee does not rule out the possibility that GDA may get stuck at suboptimal saddle points of the envelope function, which are well known to be the major challenge for training high-dimensional machine learning models (Dauphin et al., 2014; Jin et al., 2017; Zhou and Liang, 2018). On the other hand, while numerous algorithms have been developed for escaping saddle points in conventional nonconvex optimization, e.g., first-order algorithms (Ge et al., 2015; Jin et al., 2017; Carmon and Duchi, 2016; Liu and Yang, 2017) and second-order algorithms (Nesterov and Polyak, 2006; Agarwal et al., 2017; Yue et al., 2019; Zhou et al., 2018), such a type of algorithm has not been developed for escaping saddle points in nonconvex minimax optimization. Therefore, we want to ask the following fundamental questions.

- **Q:** *How to develop a provably convergent Newton-type GDA algorithm that can effectively escape saddle points in nonconvex minimax optimization? How fast it converges?*

Developing and analyzing such an algorithm is nontrivial due to the following reasons: 1) we need to have a good understanding and characterization of both the first-order and second-order information of nonconvex minimax problems; 2) we need to develop a computationally feasible and efficient GDA algorithm that can leverage the local curvature of the function to escape saddle points; 3) we aim to develop a unified analysis framework that can characterize the convergence rate of this algorithm under different types of nonconvex geometry of the minimax problem.

In this paper, we provide comprehensive answers to these questions. We develop the first Newton-type GDA algorithm that escapes strict saddle points and converges to second-order stationary points in nonconvex-strongly-concave minimax optimization. We also characterize the global and local convergence rates of this algorithm under various types of nonconvex geometry. We summarize our contributions as follows.

## 1.1 OUR CONTRIBUTIONS

We consider the minimax optimization problem (P), where $f$ is a twice-differentiable and nonconvex-strongly-concave function, and its gradient and Jacobian matrices are Lipschitz continuous. Define an envelope function $\Phi(x) := \max_{y \in \mathbb{R}^n} f(x, y)$. The existing GDA algorithms can only find first-order stationary points that satisfy $\nabla \Phi(x^*) = \mathbf{0}$. In this paper, we develop a Newton-type GDA algorithm that converges to second-order stationary points of the nonconvex minimax problem (P).

Specifically, we propose Cubic-GDA – a Newton-type GDA algorithm that leverages the classical cubic regularization technique to escape saddle points. Different from the standard cubic regularization algorithm that uses the Hessian information of the function, the Hessian of $\Phi(x)$ is not directly available in nonconvex minimax optimization, and hence we develop a rigorous and computationally feasible scheme in Cubic-GDA to estimate the Hessian.

We study the global convergence property of Cubic-GDA in general nonconvex-strongly-concave optimization. Specifically, we show that Cubic-GDA admits an intrinsic potential function $H(x, x', y)$ (see Proposition 2), which monotonically decreases along the trajectory of Cubic-GDA. Based on the monotonicity of this potential function, we show that every limit point of the parameter sequence $\{x_t\}_t$ generated by Cubic-GDA is a second-order stationary point of the minimax problem.

We further analyze the aymptotic convergence rates of Cubic-GDA under a broad spectrum of the local nonconvex Łojasiewicz gradient geometry. In this case, we show that Cubic-GDA converges to a unique limit point, which is a second-order stationary point. Moreover, as the geometry parameter increases (i.e., sharper local geometry), the convergence rate of Cubic-GDA accelerates from sub-linear convergence up to super-linear convergence, as we summarize in Table 1 below. In particular, these convergence rates are orderwise faster than those of the standard GDA under the same type of nonconvex geometry (Chen et al., 2021) [1].

---

[1] Note that the geometry parameter $\theta$ in this paper corresponds to $\frac{1}{1-\theta}$ in (Chen et al., 2021).

Table 1: Comparison of potential function value gap $H(z) - H^*$ convergence rates of Cubic-GDA and GDA under different parameterizations of Łojasiewicz gradient geometry.

| Geometry parameter | GDA (Chen et al., 2021) | **Cubic-GDA (This paper)** |
|---|---|---|
| $\theta \in (2, +\infty)$ | Super-linear convergence | Super-linear convergence |
| $\theta = 2$ | Linear convergence | Super-linear convergence |
| $\theta \in (\frac{3}{2}, 2)$ | Sub-linear convergence | Super-linear convergence |
| $\theta = \frac{3}{2}$ | Sub-linear convergence | Linear convergence |
| $\theta \in (1, \frac{3}{2})$ | Sub-linear convergence | Sub-linear convergence |

## 1.2 RELATED WORK

**Deterministic GDA algorithms:** Many studies characterized the convergence of GDA in nonconvex minimax optimization. Specifically, Lin et al. (2020); Nouiehed et al. (2019); Xu et al. (2020d) studied the convergence of GDA in the nonconvex-concave setting whereas Lin et al. (2020); Xu et al. (2020d) focused on the nonconvex-strongly-concave setting. In these general nonconvex settings, it is shown that GDA converges to a certain stationary point at a sublinear rate. Recently, Chen et al. (2021) proved the parameter convergence of proximal-GDA in regularized nonconvex-strongly-concave optimization under the Kurdyka-Łojasiewicz geometry. The convergence rates obtained there are orderwise slower than that of Cubic-GDA. Yang et al. (2020) studied an alternating gradient descent-ascent (AGDA) algorithm in which the gradient ascent step uses the current variable $x_{t+1}$ instead of $x_t$. Xu et al. (2020d) studied an alternating gradient projection algorithm which applies $\ell_2$ regularizer to the local objective function of GDA followed by projection onto the constraint sets. Daskalakis and Panageas (2018); Mokhtari et al. (2020); Zhang and Wang (2021) analyzed optimistic gradient descent-ascent (OGDA) which applies negative momentum to accelerate GDA. Mokhtari et al. (2020) also studied an extra-gradient algorithm which applies two-step GDA in each iteration. Nouiehed et al. (2019) studied multi-step GDA where multiple gradient ascent steps are performed, and they also studied the momentum-accelerated version. Cherukuri et al. (2017); Daskalakis and Panageas (2018); Jin et al. (2020) studied GDA in continuous time dynamics using differential equations. Adolphs et al. (2019) analyzed a second-order variant of the GDA algorithm. In a concurrent work (Luo and Chen, 2021), the authors proposed and studied the same Cubic-GDA algorithm. They characterize the computation complexity under a special type of inexactness that approximates the inverse Jacobian using matrix Chebyshev polynomials. As a comparison, this study focuses on analyzing the global and local convergence properties of Cubic-GDA.

**Stochastic GDA algorithms:** Lin et al. (2020); Yang et al. (2020) analyzed stochastic GDA and stochastic AGDA, which are direct extension of GDA and AGDA to the stochastic setting. Variance reduction techniques have been applied to stochastic minimax optimization, including SVRG-based (Du and Hu, 2019; Yang et al., 2020), SPIDER-based (Xu et al., 2020c), SREDA (Xu et al., 2020b), STORM (Qiu et al., 2020) and its gradient free version (Huang et al., 2020). Xie et al. (2020) studied the complexity lower bound of first-order stochastic algorithms for finite-sum minimax problem.

**Cubic regularization (CR):** CR algorithm dates back to (Griewank, 1981), where global convergence of the algorithm is established. In Nesterov and Polyak (2006), the authors analyzed the convergence rate of CR to second-order stationary points for nonconvex optimization. In (Nesterov, 2008), the authors established the sub-linear convergence of CR for solving convex smooth problems, and they further proposed an accelerated version of CR with improved sub-linear convergence. Recently, Yue et al. (2019) studied the asymptotic convergence properties of CR under the error bound condition, and established the quadratic convergence of the iterates. Recently, Hallak and Teboulle (2020) proposed a framework of two directional method for finding second-order stationary points in general smooth nonconvex optimization. This main idea of the algorithm is to search for a feasible direction toward the solution and is not based on cubic regularization. Several other works proposed different methods to solve the cubic subproblem of CR, e.g., (Agarwal et al., 2017; Carmon and Duchi, 2016; Cartis et al., 2011b). Another line of work aimed at improving the computation efficiency of CR by solving the cubic subproblem with inexact gradient and Hessian information. In particular, Ghadimi et al. (2017) proposed an inexact CR for solving convex problem. Also, Cartis et al. (2011a) proposed

an adaptive inexact CR for nonconvex optimization, whereas Jiang et al. (2017) further studied the accelerated version for convex optimization. Several studies explored subsampling schemes to implement inexact CR algorithms, e.g., (Kohler and Lucchi, 2017; Xu et al., 2020a; Zhou and Liang, 2018; Wang et al., 2018).

## 2 PROBLEM FORMULATION AND PRELIMINARIES

In this section, we introduce the problem formulation and present some preliminary results that will be used in the analysis.

**Notation:** For notation simplicity, we denote $\nabla_1 f, \nabla_2 f$ as the gradients with respect to the first and the second input arguments of $f$, respectively. We also denote $\nabla_{11} f, \nabla_{22} f$ as the Jacobian matrices where the second-order derivatives are taken over the first and second arguments of $f$, respectively. Moreover, we denote $\nabla_{12} f$ as the Jacobian matrix where the second-order derivative is taken over the first argument of $f$ and followed by the second argument, and $\nabla_{21} f$ is defined in a similar way.

We consider the minimax optimization problem (P) that satisfies the following standard assumptions.

**Assumption 1.** *The minimax optimization problem* (P) *satisfies:*

*1. Function $f(\cdot, \cdot)$ is $L_1$-smooth and function $f(x, \cdot)$ is $\mu$-strongly concave for all fixed $x$;*

*2. The Jacobian matrices $\nabla_{11} f, \nabla_{12} f, \nabla_{21} f, \nabla_{22} f$ are $L_2$-Lipschitz;*

*3. Function $\Phi$ is bounded below and has compact sub-level sets.*

To elaborate, item 1 considers the class of nonconvex-strongly-concave functions $f$ that has been widely studied in the minimax optimization literature (Lin et al., 2020; Jin et al., 2020; Xu et al., 2020d; Lu et al., 2020). Items 2 assumes that the block Jacobian matrices of $f$ are Lipschitz, which is a standard assumption for analyzing many second-order optimization algorithms (Nesterov and Polyak, 2006; Agarwal et al., 2017; Yue et al., 2019). Moreover, item 3 guarantees that the minimax problem has at least one solution. By strong concavity of $f(x, \cdot)$, it is clear that the maximizer $y^*(x) := \arg\max_{y \in \mathbb{R}^n} f(x, y)$ is unique for every $x \in \mathbb{R}^m$. In particular, if $x^*$ is a second-order stationary point of $\Phi(x)$, then $(x^*, y^*(x^*))$ is the desired solution of the minimax problem (P).

Define an envelope function $\Phi(x) := \max_{y \in \mathbb{R}^n} f(x, y)$. Then the minimax problem (P) is equivalent to the minimization problem. $\min_{x \in \mathbb{R}^m} \Phi(x)$, where $\Phi(x) = \max_{y \in \mathbb{R}^n} f(x, y)$. As we show in item 2 of Proposition 1 later, this envelope function $\Phi(x)$ is smooth and nonconvex. The existing GDA algorithms can only find first-order stationary points of the minimax problem that satisfy $\nabla \Phi(x^*) = \mathbf{0}$. In this paper, we aim to develop a provably convergent algorithm that can find second-order stationary points $x^*$ of the function $\Phi(x)$ that satisfy the following set of conditions.

$$\text{(Second-order stationary):} \quad \nabla \Phi(x^*) = \mathbf{0}, \quad \nabla^2 \Phi(x^*) \succeq \mathbf{0}.$$

In the existing literature, many optimization algorithms have been developed for finding second-order stationary points in conventional nonconvex minimization problems. This includes first-order algorithms (Ge et al., 2015; Jin et al., 2017; Carmon and Duchi, 2016; Liu and Yang, 2017) and second-order algorithms (Nesterov and Polyak, 2006; Agarwal et al., 2017; Yue et al., 2019; Zhou et al., 2018). However, these algorithms are not directly applicable to solve the problem (P'), as the function $\Phi(x)$ involves a special *maximization* structure and hence its specific function form $\Phi$ as well as the gradient $\nabla \Phi$ and Hessian $\nabla^2 \Phi$ are not available in practice. Instead, our algorithm design can only leverage information of the bi-variate function $f$.

Next, we present some important properties regarding the gradient and Jacobian matrices of the functions $f(x, y)$ and $\Phi(x)$. Throughout, we denote $\kappa = L_1 / \mu$ as the condition number.

**Proposition 1.** *Let Assumption 1 hold. Then, the following statements hold.*

*1. Mapping $y^*(x)$ is $\kappa$-Lipschitz continuous;*

*2. Function $\Phi(x)$ is $L_1(1 + \kappa)$-smooth and $\nabla \Phi(x) = \nabla_1 f(x, y^*(x))$;*

*3. Define $G(x, y) = \nabla_{11} f(x, y) - \nabla_{12} f(x, y)[\nabla_{22} f(x, y)]^{-1} \nabla_{21} f(x, y)$. Then, $G$ is a Lipschitz mapping with constant $L_G = L_2(1 + \kappa)^2$, i.e., $\|G(x', y') - G(x, y)\| \leq L_G \|(x', y') - (x, y)\|$;*

4. *The Hessian of $\Phi$ satisfies $\nabla^2\Phi(x) = G(x, y^*(x))$, which is Lipschitz continuous with constant $L_\Phi = L_G(1 + \kappa) = L_2(1 + \kappa)^3$.*

The first two items characterize the gradient of the envelope function $\Phi$ in terms of the partial gradient of the bi-variate objective function $f$. They are proved in the previous work (Lin et al., 2020) and we include them for completeness. On the other hand, the last two items further characterize the Hessian of $\Phi$ in terms of the block Jacobian matrices of $f$. As we present in the next section, the Lipschitz continuous Hessian $\nabla^2\Phi(x)$ allows us to develop a cubic-regularization-based algorithm for finding second-order stationary points. We also note that the proof of items 3 & 4 are not trivial. Specifically, we need to first develop bounds for the spectrum norm of the block Jacobian matrices in Lemma 1 (see the first page of the appendix), which helps prove the Lipschitz continuity of the $G$ mapping in item 3. Moreover, we leverage the optimality condition of $f(x, \cdot)$ to derive an expression for the maximizer mapping $y^*(x)$ (see (15) in the appendix), which is used to further prove item 4.

## 3 Cubic-GDA: Cubic-Regularized Gradient Descent-Ascent

In this section, we propose a new Gradient Descent-Ascent (GDA) algorithm that leverages the cubic regularization technique (Nesterov and Polyak, 2006) to escape strict saddle points and find second-order stationary points of the nonconvex minimax problem (P).

Our algorithm design is inspired by the conventional cubic regularization algorithm (Nesterov and Polyak, 2006). Specifically, to find a second-order stationary point of the envelope function $\Phi(x)$, the conventional cubic regularization algorithm would perform the following iterative update.

$$x_{t+1} \in \arg\min_x \nabla\Phi(x_t)^\top(x - x_t) + \frac{1}{2}(x - x_t)^\top\nabla^2\Phi(x_t)(x - x_t) + \frac{1}{6\eta_x}\|x - x_t\|^3, \quad (1)$$

where $\eta_x > 0$ is a proper learning rate. However, due to the special maximization structure of $\Phi$, its gradient and Hessian have complex formulas (see Proposition 1) that involve the mapping $y^*(x)$, which cannot be computed exactly in practice. Hence, we aim to develop a new algorithm to efficiently compute approximations of the gradient and Hessian of $\Phi$ and use them to perform the cubic regularization update.

To perform the cubic regularization update in eq. (1), we need to compute $\nabla\Phi(x_t) = \nabla_1 f(x_t, y^*(x_t))$ and $\nabla^2\Phi(x_t) = G(x_t, y^*(x_t))$ (by Proposition 1), both of which depend on the maximizer $y^*(x_t)$ of the function $f(x_t, \cdot)$. Since $f(x_t, \cdot)$ is strongly-concave, we can run $N_t$ iterations of gradient ascent to obtain an approximated maximizer $\widetilde{y}_{N_t} \approx y^*(x_t)$, and then approximate $\nabla\Phi(x_t), \nabla^2\Phi(x_t)$ using $\nabla_1 f(x_t, \widetilde{y}_{N_t})$ and $G(x_t, \widetilde{y}_{N_t})$, respectively. Intuitively, these are good approximations due to two reasons: (i) $\widetilde{y}_{N_t}$ converges to $y^*(x_t)$ at a fast linear convergence rate; and (ii) both $\nabla_1 f$ and $G$ are shown in Proposition 1 to be Lipschitz continuous in their second argument. We refer to this algorithm as **Cubic-Regularized Gradient Descent-Ascent (Cubic-GDA)**, and summarize its update rule in Algorithm 1 below.

---

**Algorithm 1** Cubic-Regularized Gradient Descent-Ascent (Cubic-GDA)

---

**Input:** Initialize $x_0, y_0$ and learning rates $\eta_x, \eta_y$.
**for** $t = 0, 1, 2, \ldots, T - 1$ **do**
    Initialize $\widetilde{y}_0 = y_t$.
    **for** $k = 0, 1, 2, \ldots, N_t - 1$ **do**
        $\widetilde{y}_{k+1} = \widetilde{y}_k + \eta_y\nabla_2 f(x_t, \widetilde{y}_k)$.
    **end**
    Set $y_{t+1} = \widetilde{y}_{N_t}$ and compute $G(x_t, y_{t+1})$ as follows:

    $G(x_t, y_{t+1}) = \nabla_{11}f(x_t, y_{t+1}) - \nabla_{12}f(x_t, y_{t+1})[\nabla_{22}f(x_t, y_{t+1})]^{-1}\nabla_{21}f(x_t, y_{t+1})$.

    $x_{t+1} \in \arg\min_x \nabla_1 f(x_t, y_{t+1})^\top(x - x_t) + \frac{1}{2}(x - x_t)^\top G(x_t, y_{t+1})(x - x_t) + \frac{1}{6\eta_x}\|x - x_t\|^3$.
**end**
**Output:** $x_T, y_T$.

---

We further comment on the implementation of Cubic-GDA. We note that the Cubic-GDA updates in Algorithm 1 can be implemented in a computation efficient way. First, note that Cubic-GDA involves

an inner loop that performs gradient ascent updates. To guarantee the convergence of the algorithm, we prove in the next section that the number of inner iterations $N_t$ only needs to be kept at logarithm scale. Therefore, a few number of inner iterations suffice to guarantee convergence in practice. Second, the cubic regularization sub-problem can be efficiently solved by the gradient descent algorithm (Carmon and Duchi, 2016), and it involves computation of only *Jacobian-vector product* that can be efficiently computed by the existing machine learning platforms such as TensorFlow (Abadi et al., 2015) and PyTorch (Paszke et al., 2019). In particular, to compute the Hessian-vector product $G(x, y)v$ for any vector $v$, one needs to compute the Jacobian-vector product $\nabla_{11}f(x, y)v$ and the matrix-vector product $\nabla_{12}f(x, y)[\nabla_{22}f(x, y)]^{-1}\nabla_{21}f(x, y)v$. We note that this matrix-vector product term can be computed as follows: first compute the Jacobian-vector product $b = \nabla_{21}f(x, y)v$; Then, solve the invertible linear system $\nabla_{22}f(x, y)u = b$ using any standard solver (e.g., conjugate gradient method), which involves iteratively computing Jacobian-vector products $\nabla_{22}f(x, y)w$ for some vector $w$; Finally, compute the Jacobian-vector product $\nabla_{12}f(x, y)u$. Hence, one can call multiple Jacobian-vector product oracles to solve the cubic regularization sub-problem.

## 4 GLOBAL CONVERGENCE PROPERTIES OF CUBIC-GDA

In this section, we study the global convergence properties of Cubic-GDA. Importantly, our analysis is based on characterizing an intrinsic potential function of the Cubic-GDA algorithm in nonconvex minimax optimization.

Recall that our goal is to find a second-order stationary point of the function $\Phi(x)$. Our next result shows that Cubic-GDA admits an intrinsic potential function that monotonically decreases in the optimization process. The proof of is included in Appendix B.

**Proposition 2.** *Let Assumption 1 hold. Define the following potential function*

$$H(x, x', y) := \Phi(x) + L_2\kappa^3\|x' - x\|^3 + 4L_2\|y - y^*(x)\|^3,$$

*and denote $H_t := H(x_t, x_{t-1}, y_{t+1})$. Choose $N_t \geq \frac{\max(\ln 2, \ln[L_1\|\nabla_2 f(x_t, y_t)\|/(L_2\mu)] - 2\ln\|x_t - x_{t-1}\|)}{\ln[\kappa/(\kappa-1)]}$ and learning rates $\eta_x \leq \frac{1}{28L_2\kappa^3}$, $\eta_y \leq \frac{2}{L+\mu}$. Then, the sequences $\{x_t, y_t\}_t$ generated by Cubic-GDA satisfy, for all $t = 0, 1, 2, ...$*

$$H_{t+1} \leq H_t - L_2\kappa^3\|x_{t+1} - x_t\|^3 - L_2\|y_{t+1} - y^*(x_t)\|^3. \tag{2}$$

*Consequently, it holds that*

$$\lim_{t\to\infty}\|x_{t+1} - x_t\| = 0, \quad \lim_{t\to\infty}\|y_{t+1} - y_t\| = 0, \quad \lim_{t\to\infty}\|y_t - y^*(x_t)\| = 0.$$

**Remark 1.** *We note that the above key result can also be established for an inexact version of Cubic-GDA, which formulates the cubic subproblem with a general inexact gradient $p_t \approx \nabla_1 f(x_t, y_{t+1})$ and inexact Jacobian $P_t \approx G(x_t, y_{t+1})$ that satisfy the conditions*

$$\|p_t - \nabla_1 f(x_t, y_{t+1})\| \leq O(\|x_{t+1} - x_t\|^2), \quad \|P_t - G(x_t, y_{t+1})\| \leq O(\|x_{t+1} - x_t\|).$$

*These inexactness conditions are widely studied in the existing literature Cartis et al. (2011b;a). Under these inexact conditions, our proof of the above proposition remains unchanged, except that the coefficients of the term $\|x_{t+1} - x_t\|^3$ would be slightly different.*

Proposition 2 reveals that Cubic-GDA admits an intrinsic potential function $H$, which is the objective function $\Phi(x)$ regularized by two cubic terms $\|x' - x\|^3$, $\|y - y^*(x)\|^3$. Such a potential function is closely connected to the optimization goal. Specifically, consider a desired case where $x_t$ converges to a certain second-order stationary point $x^*$ and $y_t$ converges to $y^*(x^*)$, it is clear that the potential function $H_t$ would converge to the corresponding function value $\Phi(x^*)$. Hence, minimizing the function $\Phi$ is equivalent to minimizing the potential function $H$. More importantly, Proposition 2 shows that this potential function is monotonically decreasing along the optimization path of Cubic-GDA, implying that the algorithm continuously makes optimization progress. By leveraging this property of the potential function, we are able to show that the parameter sequences generated by Cubic-GDA are asymptotically stable, i.e., $x_{t+1} - x_t \to \mathbf{0}, y_t \to y^*(x_t)$.

**Remark 2.** *In each outer iteration $t$, we set the total number of inner gradient ascent iterations $N_t$ based on $\|\nabla_2 f(x_t, y_t)\|$ and $\|x_t - x_{t-1}\|$. Note that both of these two quantities can be easily computed right after iteration $t - 1$.*

Based on Proposition 2, we are able to prove the convergence of $\{x_t\}_t$ to a certain second-order stationary point, which we formally present in the next theorem. The proof is included in Appendix C.

**Theorem 1** (Global convergence). *Under the same conditions as those of Proposition 2, the Cubic-GDA has the following global convergence properties.*

1. *The function value sequence $\{\Phi(x_t)\}_t$ converges to a finite limit $H^* > -\infty$;*

2. *The generated sequences $\{x_t\}_t, \{y_t\}_t$ are bounded and have a compact sets of limit points.*

3. *Every limit point $x^*$ of $\{x_t\}_t$ is a second-order stationary point of $\Phi$, i.e., $\nabla \Phi(x^*) = \mathbf{0}$, $\nabla^2 \Phi(x^*) \succeq \mathbf{0}$, and satisfies $\Phi(x^*) = H^*$.*

The above theorem establishes the global convergence properties of Cubic-GDA. Specifically, item 1 shows that the function value sequence $\{\Phi(x_t)\}_t$ converges to a finite limit $H^*$, which is also the limit of the potential function sequence $\{H_t\}_t$. Moreover, items 2 & 3 further show that all the limit points of $\{x_t\}_t$ are second-order stationary points of the minimax problem, at which the function $\Phi$ achieves the constant value $H^*$. These results show that Cubic-GDA is guaranteed to find second-order stationary points in nonconvex minimax optimization.

By further leveraging the potential function characterized in Proposition 2, we obtain the following global convergence rate of Cubic-GDA to a second-order stationary point. The proof is included in Appendix D. Throughout, we adopt the following standard measure of second-order stationary introduced in (Nesterov and Polyak, 2006).

$$\mu(x) = \max \left\{ \sqrt{\frac{\|\nabla \Phi(x)\|}{1/(2\eta_x) + 5L_2\kappa^3 + 4L_2^2\kappa^2/L_1}}, \frac{-\lambda_{\min}(\nabla^2 \Phi(x))}{1/(2\eta_x) + 4L_2^2\kappa^2/L_1} \right\}.$$

Intuitively, a smaller $\mu(x)$ means that the point $x$ is closer to being second-order stationary.

**Theorem 2** (Global convergence rate). *Under the same conditions as those of Proposition 2, the Cubic-GDA converges at the following rate for all $T \geq \frac{H_0 - \inf_{x \in \mathbb{R}^m} \Phi(x)}{L_2\kappa^3/3}$.*

$$\min_{0 \leq t \leq T-1} \mu(x_t) \leq \left( \frac{H_0 - \inf_{x \in \mathbb{R}^m} \Phi(x)}{TL_2\kappa^3/3} \right)^{1/3}.$$

The above theorem shows that the first-order stationary measure $\|\nabla \Phi(x_t)\|$ converges at a sublinear rate $\mathcal{O}(T^{-\frac{2}{3}})$, and the second-order stationary measure $-\lambda_{\min}(\nabla^2 \Phi(x))$ converges at a sublinear rate $\mathcal{O}(T^{-\frac{1}{3}})$. Both results match the convergence rates of the cubic regularization algorithm for nonconvex minimization (Nesterov and Polyak, 2006). Therefore, by leveraging the curvature of the approximated Hessian matrix $G(x_t, y_{t+1})$, Cubic-GDA is able to escape strict saddle points of $\Phi$ at a fast rate.

We note that the proof of the global convergence results in Theorems 1 and 2 are critically based on the intrinsic potential function $H$ that we characterized in Proposition 2. We elaborate our technical contribution as follows.

- First, to identify the potential function, we need to characterize the per-iteration progress induced by the cubic regularization step. However, the cubic subproblem in Cubic-GDA is constructed by an inexact gradient $\nabla_1 f(x_t, y_{t+1})$ and Hessian matrix $G(x_t, y_{t+1})$. Therefore, we need to properly choose the number of inner gradient ascent iterations $N_t$ to control the estimation error of both the gradient and Hessian approximations at a desired level.

- Due to the inexactness of the gradient and Hessian matrix, the cubic regularization update of Cubic-GDA does not lead to a monotonically decreasing function value $\Phi(x_t)$, as opposed to the original cubic regularization algorithm in nonconvex minimization (which uses exact gradient and Hessian). Hence, we construct and identify a decreasing potential function $H$ instead.

## 5 CONVERGENCE ANALYSIS UNDER LOCAL NONCONVEX GEOMETRY

The (2) of Proposition 2 shows that Cubic-GDA has a special optimization dynamics and therefore its convergence rate is expected to be different from that of the vanilla GDA in nonconvex minimax

optimization. In this section, we explore the convergence rates of Cubic-GDA under a broad spectrum of local nonconvex geometries characterized by the Łojasiewicz gradient inequality.

We first introduce the Łojasiewicz gradient geometry of a function $h$. Throughout, the point-to-set distance is denoted as $\text{dist}_\Omega(x) := \inf_{u \in \Omega} \|x - u\|$.

**Definition 1.** *A differentiable function $h$ is said to satisfy the Łojasiewicz gradient geometry if for every compact set $\Omega$ of critical points on which $h$ takes a constant value $h_\Omega \in \mathbb{R}$, there exist $\varepsilon, \lambda > 0$ such that for all $x \in \{z \in \mathbb{R}^m : \text{dist}_\Omega(z) < \varepsilon, h_\Omega < h(z) < h_\Omega + \lambda\}$, the following condition holds.*

$$h(x) - h_\Omega \le c\|\nabla h(x)\|^\theta, \tag{3}$$

*where $c > 0$ is a universal constant and $\theta \in (1, +\infty)$ is the geometry parameter.*

Intuitively, the Łojasiewicz gradient geometry is a gradient-dominant-type geometry that characterizes the local geometry of a nonconvex function around the set of critical points. In particular, it generalizes the Polyak-Łojasiewicz (PL) geometry that corresponds to the special case $\theta = 2$ Łojasiewicz (1963); Karimi et al. (2016). In fact, a generalized version of the Łojasiewicz gradient geometry has been shown to hold for a large class of functions including sub-analytic functions, exponential functions and semi-algebraic functions, which cover most of the nonconvex functions encountered in machine learning applications (Zhou et al., 2016; Yue et al., 2019; Zhou and Liang, 2017; Zhou et al., 2018). For example, consider the class of robust machine learning problems that involve the minimax problem $\min_\theta \max_{\xi_i} \frac{1}{n} \sum_{i=1}^n \ell(h_\theta(\xi_i), y_i)^2 - \frac{\lambda}{2}\|\xi_i - a_i\|^2$. Here $(x_i, y_i)$ is the $i$-th data sample that includes, e.g., an image $x_i$ and its label $y_i$, $\xi_i$ denotes the adversarial image, $h_\theta$ is a classification model parameterized by $\theta$, and $\ell$ denotes the loss function. Such a problem is nonconvex-stongly-concave when $\lambda$ is large. In particular, as elaborated in the appendix of (Bolte et al., 2014), the envelop function $\Phi(x) := \max_{\xi_i} \frac{1}{n} \sum_{i=1}^n \ell(h_\theta(\xi_i), y_i)^2 - \frac{\lambda}{2}\|\xi_i - a_i\|^2$ satisfies the local Lojasiewicz gradient geometry if it is semi-algebraic, which holds if every sample loss $f(\theta, \xi_i) := \ell(h_\theta(\xi_i), y_i)^2 - \frac{\lambda}{2}\|\xi_i - a_i\|^2$ is semi-algebraic.

By (2) of Proposition 2 and (30) (proved in Appendix E), we show that the potential function $H$ of Cubic-GDA satisfies the following special optimization dynamics.

$$H_{t+1} - H_t \le -\mathcal{O}\big(\|x_{t+1} - x_t\|^3\big), \tag{4}$$

$$\|\nabla H_t\| \le \mathcal{O}\big(\|x_t - x_{t-1}\|^2 + \|x_{t-1} - x_{t-2}\|^2\big). \tag{5}$$

The above dynamics of Cubic-GDA involves higher-order terms than the dynamics of GDA, which takes the form $H_{t+1} - H_t \le -\mathcal{O}(\|x_{t+1} - x_t\|^2)$ and $\|\nabla H_t\| \le \mathcal{O}(\|x_{t+1} - x_t\|)$ (Chen et al., 2021). Thus, it is expected that Cubic-GDA achieves a faster convergence rate than GDA. On the other hand, compare with the dynamics of the cubic regularization algorithm (Zhou et al., 2018), the gradient dynamic of Cubic-GDA in (5) involves an additional term $\|x_{t-1} - x_{t-2}\|^2$ that depends on the history, which is due to the inexact gradient and Hessian used in the cubic regularization step.

With the optimization dynamics in (4) and (5) and by leveraging the Łojasiewicz gradient geometry, we are able to prove the following strengthened convergence result of Cubic-GDA.

**Theorem 3.** *Let Assumption 1 hold and assume that the potential function $H$ satisfies the local Łojasiewicz gradient geometry. Choose the hyperparameters $N_t, \eta_x, \eta_y$ in the same way as Proposition 2. Then, the sequences $\{(x_t, y_t)\}_t$ generated by Cubic-GDA have a unique limit point, which is a second-order stationary point of $\Phi$.*

Recall that Theorem 1 only proves that every limit point of $\{x_t\}_t$ is a second-order stationary point of the minimax problem. Theorem 3 further strengthens Theorem 1 by showing that Cubic-GDA converges to a unique second-order stationary limit point under the Łojasiewicz gradient geometry.

Next, we further study the asymptotic convergence rates of Cubic-GDA under different parameter ranges of the Łojasiewicz gradient geometry. We first obtain the following function value convergence rate result, which strengthens the convergence rate result established in Theorem 2. Throughout, $t_0$ denotes a sufficiently large positive integer and $C_0$ is a universal positive constant defined as

$$C_0 = \sqrt[3]{2}c^{1/\theta}L_2^{-2/3}\Big(10L_2\kappa + \frac{24L_2^3}{L_1^2\kappa} + \frac{4L_2^2}{L_1} + \frac{1}{2\eta_x\kappa^2}\Big)$$

The proof is included in Appendix F.

**Theorem 4** (Funtion value convergence rate). *Under the same conditions as those of Theorem 3, the sequence of potential function $\{H_t\}_t$ converges to the limit $H^*$ at the following rates.*

1. *If the geometry parameter $\theta \in (\frac{3}{2}, \infty)$, then $H_t \downarrow H^*$ super-linearly as*

$$H_t - H^* \leq \mathcal{O}\Big( \exp\Big( -\Big(\frac{2\theta}{3}\Big)^{\frac{t-t_0}{2}} \Big) \Big), \quad \forall t \geq t_0; \tag{6}$$

2. *If the geometry parameter $\theta = \frac{3}{2}$, then $H_t \downarrow H^*$ linearly as*

$$H_t - H^* \leq (1 + C_0^{3/2})^{-\frac{t-t_0}{2}}, \quad \forall t \geq t_0; \tag{7}$$

3. *If the geometry parameter $\theta \in (1, \frac{3}{2})$, then $H_t \downarrow H^*$ sub-linearly as*

$$H_t - H^* \leq \mathcal{O}\Big( (t - t_0)^{-\frac{2\theta}{3-2\theta}} \Big), \quad \forall t \geq t_0. \tag{8}$$

**Remark 3.** *We note that if the Łojasiewicz gradient geometry holds globally, then the above asymptotic convergence rates become global convergence rates.*

The above theorem characterizes the convergence rates of the potential function of Cubic-GDA in the full spectrum of $\theta$ of the local Łojasiewicz gradient geometry. Specifically, it shows that a larger $\theta$ implies that the local geometry is sharper and hence leads to a faster convergence rate. In particular, as we summarize in Table 1 in the introduction section, the convergence rate of Cubic-GDA is orderwise faster than that of the vanilla GDA for a wide range of the parameter of the Łojasiewicz gradient geometry. This demonstrates the advantage of leveraging higher-order information in nonconvex minimax-optimization.

As a byproduct, we also obtain the following asymptotic convergence rates of the parameter sequences generated by Cubic-GDA under the Łojasiewicz gradient geometry. The proof is included in Appendix G.

**Theorem 5** (Parameter convergence rate). *Under the same conditions as those of Theorem 3, the sequences $\{x_t, y_t\}_t$ generated by Cubic-GDA converge to their limits $x^*, y^*(x^*)$ respectively at the following rates.*

1. *If the geometry parameter $\theta \in (\frac{3}{2}, \infty)$, then $(x_t, y_t) \to (x^*, y^*(x^*))$ super-linearly as*

$$\max\big\{ \|x_t - x^*\|, \|y_t - y^*(x^*)\| \big\} \leq \mathcal{O}\Big( \exp\Big( -\frac{1}{3}\Big(\frac{2\theta}{3}\Big)^{\frac{t-t_0}{2}-1} \Big) \Big), \quad \forall t \geq t_0; \tag{9}$$

2. *If the geometry parameter $\theta = \frac{3}{2}$, then $(x_t, y_t) \to (x^*, y^*(x^*))$ linearly as*

$$\max\big\{ \|x_t - x^*\|, \|y_t - y^*(x^*)\| \big\} \leq \mathcal{O}\Big( (1 + C_0^{3/2})^{-\frac{t-t_0}{2}} \Big), \quad \forall t \geq t_0; \tag{10}$$

3. *If the geometry parameter $\theta \in (1, \frac{3}{2})$, then $(x_t, y_t) \to (x^*, y^*(x^*))$ sub-linearly as*

$$\|x_t - x^*\| \leq \mathcal{O}\Big( (t - t_0)^{-\frac{2(\theta-1)}{3-2\theta}} \Big), \ \|y_t - y^*(x_t)\| \leq \mathcal{O}\Big( (t - t_0)^{-\frac{2\theta}{3(3-2\theta)}} \Big), \quad \forall t \geq t_0. \tag{11}$$

It can be seen that, similar to the convergence rate results of the function value sequence, the convergence rate of the parameter sequence is also affected by the parameterization of the local geometry.

# 6 CONCLUSION

In this paper, we take one step further toward improving the convergence guarantee of GDA-type algorithms in nonconvex minimax optimization. We develop a Cubic-GDA algorithm that leverages the second-order information and the cubic regularization technique to effectively escape strict saddle points in nonconvex minimax optimization. Our key observation is that Cubic-GDA has an intrinsic potential function that monotonically decreases in the optimization process, and this leads to a guaranteed global convergence of the algorithm. Moreover, our convergence analysis shows that Cubic-GDA achieves a faster convergence rate than the standard GDA for a wide spectrum of gradient dominant-type nonconvex geometries. In the future study, we will develop stochastic variants of Cubic-GDA to further improve its computation efficiency. Another interesting direction is to apply momentum techniques to further accelerate the convergence of this algorithm.

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

# Appendix

## Table of Contents

In this supplementary material, we present the proof of all the results claimed in the paper. We first prove the following auxiliary lemma that bounds the spectral norm of the Jacobian matrices.

**Lemma 1.** *Let Assumption 1 hold. Then, for any $x \in \mathbb{R}^m$ and $y \in \mathbb{R}^n$, the Jacobian matrices of $f(x, y)$ satisfy the following bounds.*

$$\|[\nabla_{22}f(x,y)]^{-1}\| \leq \mu^{-1}, \tag{12}$$

$$\|\nabla_{12}f(x,y)\| = \|\nabla_{21}f(x,y)\| \leq L_1. \tag{13}$$

*Proof.* We first prove eq. (12). Consider any $x \in \mathbb{R}^m$ and $y \in \mathbb{R}^n$. By Assumption 1 we know that $f(x, \cdot)$ is $\mu$-strongy concave, which implies that $-\nabla_{22}f(x,y) \succeq \mu I$. Thus, we further conclude that

$$\|[\nabla_{22}f(x,y)]^{-1}\| = \lambda_{\max}\big([-\nabla_{22}f(x,y)]^{-1}\big) = \Big(\lambda_{\min}\big(-\nabla_{22}f(x,y)\big)\Big)^{-1} \leq \mu^{-1}.$$

Next, we prove eq. (13). Consider any $x, u \in \mathbb{R}^m$ and $y \in \mathbb{R}^n$, we have

$$\begin{aligned}
\|\nabla_{21}f(x,y)u\| &= \Big\|\frac{\partial}{\partial t}\nabla_2 f(x+tu,y)\Big|_{t=0}\Big\| \\
&= \Big\|\lim_{t\to 0}\frac{1}{t}\big[\nabla_2 f(x+tu,y) - \nabla_2 f(x,y)\big]\Big\| \\
&= \lim_{t\to 0}\frac{1}{|t|}\big\|\nabla_2 f(x+tu,y) - \nabla_2 f(x,y)\big\| \\
&\leq \lim_{t\to 0}\frac{L_1}{|t|}\big\|tu\big\| = L_1\|u\|, \tag{14}
\end{aligned}$$

which implies that $\|\nabla_{21}f(x,y)\| \leq L_1$. Since $f$ is twice differentiable and has continuous second-order derivative, we have $\nabla_{12}f(x,y)^\top = \nabla_{21}f(x,y)$, and hence eq. (13) follows. $\qquad\square$

## A    PROOF OF PROPOSITION 1

**Proposition 1.** *Let Assumption 1 hold. Then, the following statements hold.*

*1. Mapping $y^*(x)$ is $\kappa$-Lipschitz continuous;*

*2. Function $\Phi(x)$ is $L_1(1+\kappa)$-smooth and $\nabla\Phi(x) = \nabla_1 f(x, y^*(x))$;*

*3. Define $G(x,y) = \nabla_{11}f(x,y) - \nabla_{12}f(x,y)[\nabla_{22}f(x,y)]^{-1}\nabla_{21}f(x,y)$. Then, $G$ is a Lipschitz mapping with constant $L_G = L_2(1+\kappa)^2$, i.e., $\|G(x',y') - G(x,y)\| \leq L_G\|(x',y') - (x,y)\|$;*

4. *The Hessian of $\Phi$ satisfies $\nabla^2\Phi(x) = G(x, y^*(x))$, which is Lipschitz continuous with constant $L_\Phi = L_G(1+\kappa) = L_2(1+\kappa)^3$.*

*Proof.* The items 1 & 2 are proved in Chen et al. (2021); Lin et al. (2020).

We first prove the item 3. Consider any $x, x' \in \mathbb{R}^m$ and $y, y' \in \mathbb{R}^n$. For convenience we denote $z = (x, y)$ and $z' = (x', y')$. Then, by Assumption 1 and using the bounds of Lemma 1, we have that

$$
\begin{aligned}
&\|G(x', y') - G(x, y)\| \\
&\leq \|\nabla_{11}f(x', y') - \nabla_{11}f(x, y)\| + \|\nabla_{12}f(x', y') - \nabla_{12}f(x, y)\|\|[\nabla_{22}f(x', y')]^{-1}\|\|\nabla_{21}f(x', y')\| \\
&\quad + \|\nabla_{12}f(x, y)\|\|[\nabla_{22}f(x', y')]^{-1} - [\nabla_{22}f(x, y)]^{-1}\|\|\nabla_{21}f(x', y')\| \\
&\quad + \|\nabla_{12}f(x, y)\|\|[\nabla_{22}f(x, y)^{-1}]\|\|\nabla_{21}f(x', y') - \nabla_{21}f(x, y)\| \\
&\leq L_2\|z' - z\| + (L_2\|z' - z\|)\mu^{-1}L_1 \\
&\quad + L_1^2\|[\nabla_{22}f(x', y')]^{-1}\|\|\nabla_{22}f(x, y) - \nabla_{22}f(x', y')\|\|[\nabla_{22}f(x, y)]^{-1}\| + L_1\mu^{-1}(L_2\|z' - z\|) \\
&\leq L_2(1 + 2\kappa)\|z' - z\| + L_1^2\mu^{-1}(L_2\|z' - z\|)\mu^{-1} \\
&\leq L_2(1 + \kappa)^2\|z' - z\|.
\end{aligned}
$$

Next, we prove the item 4. Consider any fixed $x \in \mathbb{R}^m$, we know that $f(x, \cdot)$ achieves its maximum at $y^*(x)$, where the gradient vanishes, i.e., $\nabla_2 f(x, y^*(x)) = \mathbf{0}$. Thus, we further obtain that

$$\mathbf{0} = \nabla_x\nabla_2 f(x, y^*(x)) = \nabla_{21}f(x, y^*(x)) + \nabla_{22}f(x, y^*(x))\nabla y^*(x),$$

which implies that

$$\nabla y^*(x) = -[\nabla_{22}f(x, y^*(x))]^{-1}\nabla_{21}f(x, y^*(x)). \tag{15}$$

With the above equation, we take derivative of $\nabla\Phi(x) = \nabla_1 f(x, y^*(x))$ and obtain that

$$
\begin{aligned}
\nabla^2\Phi(x) &= \nabla_{11}f(x, y^*(x)) + \nabla_{12}f(x, y^*(x))\nabla y^*(x) \\
&= \nabla_{11}f(x, y^*(x)) - \nabla_{12}f(x, y^*(x))[\nabla_{22}f(x, y^*(x))]^{-1}\nabla_{21}f(x, y^*(x)) \\
&= G(x, y^*(x)).
\end{aligned} \tag{16}
$$

Moreover, we have that

$$
\begin{aligned}
\|\nabla^2\Phi(x') - \nabla^2\Phi(x)\| &= \|G(x', y^*(x')) - G(x, y^*(x))\| \\
&\leq L_G\big[\|x' - x\| + \|y^*(x') - y^*(x)\|\big] \\
&\leq L_G(1 + \kappa)\|x' - x\|,
\end{aligned} \tag{17}
$$

where the last step uses the item 1. This proves the item 4.

$\square$

# B    PROOF OF PROPOSITION 2

**Proposition 2.** *Let Assumption 1 hold. Define the following potential function*

$$H(x, x', y) := \Phi(x) + L_2\kappa^3\|x' - x\|^3 + 4L_2\|y - y^*(x)\|^3,$$

*and denote $H_t := H(x_t, x_{t-1}, y_{t+1})$. Choose $N_t \geq \frac{\max(\ln 2, \ln[L_1\|\nabla_2 f(x_t, y_t)\|/(L_2\mu)] - 2\ln\|x_t - x_{t-1}\|)}{\ln[\kappa/(\kappa-1)]}$ and learning rates $\eta_x \leq \frac{1}{28L_2\kappa^3}$, $\eta_y \leq \frac{2}{L+\mu}$. Then, the sequences $\{x_t, y_t\}_t$ generated by Cubic-GDA satisfy, for all $t = 0, 1, 2, ...$*

$$H_{t+1} \leq H_t - L_2\kappa^3\|x_{t+1} - x_t\|^3 - L_2\|y_{t+1} - y^*(x_t)\|^3. \tag{2}$$

*Consequently, it holds that*

$$\lim_{t\to\infty}\|x_{t+1} - x_t\| = 0, \quad \lim_{t\to\infty}\|y_{t+1} - y_t\| = 0, \quad \lim_{t\to\infty}\|y_t - y^*(x_t)\| = 0.$$

*Proof.* We first bound the term $\|y_{t+1} - y^*(x_t)\|$, which corresponds to the optimality gap of the maximization problem. Note that $y^*(x_t) \in \mathbb{R}^n$ is the unique maximizer of the strongly concave function $f(x_t, y)$. Note that $y_{t+1}$ is obtained by applying $N_t$ gradient ascent steps starting from $y_t$. Hence, By the convergence rate of gradient ascent algorithm under strong concavity, we conclude that with learning rate $\eta_y = \frac{2}{L+\mu}$,

$$\|y_{t+1} - y^*(x_t)\| \leq (1 - \kappa^{-1})^{N_t} \|y_t - y^*(x_t)\| \tag{18}$$

$$\leq \frac{L_2}{L_1} \|x_t - x_{t-1}\|^2, \tag{19}$$

where the second inequality uses the hyperparameter choice

$$N_t \geq \frac{\ln[L_1 \|\nabla_2 f(x_t, y_t)\|/(L_2 \mu)] - 2\ln\|x_t - x_{t-1}\|}{\ln[\kappa/(\kappa-1)]} \geq \frac{\ln[L_1 \|y_t - y^*(x_t)\|/L_2] - 2\ln\|x_t - x_{t-1}\|}{\ln[\kappa/(\kappa-1)]},$$

as $\|\nabla_2 f(x_t, y_t)\| = \|\nabla_2 f(x_t, y_t) - \nabla_2 f(x_t, y^*(x_t))\| \geq \mu\|y_t - y^*(x_t)\|$. Moreover, eq. (18) and the hyperparameter choice that $N_t \geq \frac{\ln 2}{\ln[\kappa/(\kappa-1)]}$ imply that

$$\|y_{t+2} - y^*(x_{t+1})\|^3 \leq (1 - \kappa^{-1})^{3N_t} \|y_{t+1} - y^*(x_{t+1})\|^3$$

$$\overset{(i)}{\leq} \left\|\frac{1}{2}a + \frac{1}{2}b\right\|^3$$

$$\overset{(ii)}{\leq} \frac{1}{2}\|y_{t+1} - y^*(x_t)\|^3 + \frac{\kappa^3}{2}\|x_{t+1} - x_t\|^3. \tag{20}$$

where (i) denotes that $a := y_{t+1} - y^*(x_t)$, $b := y^*(x_t) - y^*(x_{t+1})$, (ii) applies Jensen's inequality to the convex function $\|\cdot\|^3$.

Based on Theorem 10 and Proposition 1 of Nesterov and Polyak (2006), we know that

$$\lambda_{\min}\left[G(x_t, y_{t+1})\right] \geq -\frac{1}{2\eta_x}\|x_{t+1} - x_t\|. \tag{21}$$

Since $x_{t+1}$ minimizes the following cubic sub-problem

$$g_{y_{t+1}}(x) := \nabla_1 f(x_t, y_{t+1})^\top (x - x_t) + \frac{1}{2}(x - x_t)^\top G(x_t, y_{t+1})(x - x_t) + \frac{1}{6\eta_x}\|x - x_t\|^3,$$

we obtain the following optimality condition

$$\nabla_1 f(x_t, y_{t+1}) + G(x_t, y_{t+1})(x_{t+1} - x_t) + \frac{1}{2\eta_x}\|x_{t+1} - x_t\|(x_{t+1} - x_t) = \mathbf{0}. \tag{22}$$

Next, by the Lipschitz Hessian of $\Phi$, we obtain that

$\Phi(x_{t+1}) - \Phi(x_t)$

$$\leq \left(\nabla\Phi(x_t) - \nabla_1 f(x_t, y_{t+1})\right)^\top (x_{t+1} - x_t) + \frac{1}{2}(x_{t+1} - x_t)^\top \left(\nabla^2\Phi(x_t) - G(x_t, y_{t+1})\right)(x_{t+1} - x_t)$$

$$+ \nabla_1 f(x_t, y_{t+1})^\top (x_{t+1} - x_t) + \frac{1}{2}(x_{t+1} - x_t)^\top G(x_t, y_{t+1})(x_{t+1} - x_t) + \frac{L_\Phi}{6}\|x_{t+1} - x_t\|^3$$

$$\overset{(i)}{\leq} L_1\|y_{t+1} - y^*(x_t)\|\|x_{t+1} - x_t\| + \frac{L_G}{2}\|x_{t+1} - x_t\|^2\|y_{t+1} - y^*(x_t)\|$$

$$- \frac{1}{2}(x_{t+1} - x_t)^\top G(x_t, y_{t+1})(x_{t+1} - x_t) + \left(\frac{L_\Phi}{6} - \frac{1}{2\eta_x}\right)\|x_{t+1} - x_t\|^3$$

$$\overset{(ii)}{\leq} \frac{2L_1^{3/2}}{3L_2^{1/2}\kappa^{3/2}}\|y_{t+1} - y^*(x_t)\|^{3/2} + \frac{L_2\kappa^3}{3}\|x_{t+1} - x_t\|^3$$

$$+ \frac{L_G}{2}\left[\frac{2\kappa}{3}\|x_{t+1} - x_t\|^3 + \frac{1}{3\kappa^2}\|y_{t+1} - y^*(x_t)\|^3\right]$$

$$+ \frac{1}{4\eta_x}\|x_{t+1} - x_t\|^3 + \left(\frac{L_\Phi}{6} - \frac{1}{2\eta_x}\right)\|x_{t+1} - x_t\|^3 \tag{23}$$

$$\overset{(iii)}{\leq} \frac{2L_2}{3\kappa^{3/2}}\|x_t - x_{t-1}\|^3 + \frac{2L_2}{3}\|y_{t+1} - y^*(x_t)\|^3 + \left(3L_2\kappa^3 - \frac{1}{4\eta_x}\right)\|x_{t+1} - x_t\|^3, \tag{24}$$

where (i) uses Assumption 1, the item 4 of Proposition 1 and eq. (22), (ii) uses eq. (21) and the inequality that $ab = ((Ca)^{3/2}(Ca)^{3/2}(b/C)^3)^{1/3} \leq \frac{2}{3}(Ca)^{3/2} + \frac{b^3}{3C^3}$ for any $a, b \geq 0$ and $C > 0$ (based on AM-GM inequality), (iii) uses eq. (19) and $L_G = L_2(1+\kappa)^2 \leq 4L_2\kappa^2$, $L_\Phi = L_2(1+\kappa)^3 \leq 8L_2\kappa^3$.

Multiplying eq. (20) with $4L_2$ and adding it to eq. (24) yield that

$$\Phi(x_{t+1}) - \Phi(x_t) + 4L_2\|y_{t+2} - y^*(x_{t+1})\|^3$$

$$\leq \frac{2L_2}{3\kappa^{3/2}}\|x_t - x_{t-1}\|^3 + \frac{8L_2}{3}\|y_{t+1} - y^*(x_t)\|^3 + \left(5L_2\kappa^3 - \frac{1}{4\eta_x}\right)\|x_{t+1} - x_t\|^3,$$

$$\overset{(i)}{\leq} L_2\kappa^3\|x_t - x_{t-1}\|^3 + 3L_2\|y_{t+1} - y^*(x_t)\|^3 - 2L_2\kappa^3\|x_{t+1} - x_t\|^3,$$

where (i) uses the condition that $\eta_x \leq \frac{1}{28L_2\kappa^3}$. The above inequality implies eq. (2) by defining $H_t := \Phi(x_t) + L_2\kappa^3\|x_t - x_{t-1}\|^3 + 4L_2\|y_{t+1} - y^*(x_t)\|^3$.

Next, summing eq. (2) over $t = 0, 1, ..., T-1$, we obtain that for all $T \geq 1$,

$$L_2\kappa^3 \sum_{t=0}^{T-1} \|x_{t+1} - x_t\|^3 + L_2 \sum_{t=0}^{T-1} \|y_{t+1} - y^*(x_t)\|^3$$

$$\leq H_0 - H_T \leq H_0 - \Phi(x_T) \leq H_0 - \inf_{x \in \mathbb{R}^m} \Phi(x) < +\infty. \tag{25}$$

Letting $T \to \infty$ yields that $\sum_{t=0}^{\infty} \|x_{t+1} - x_t\|^3 < +\infty$ and $\sum_{t=0}^{\infty} \|y_{t+1} - y^*(x_t)\|^3 < +\infty$, so $\lim_{t\to\infty} \|x_{t+1} - x_t\| = \lim_{t\to\infty} \|y_{t+1} - y^*(x_t)\| = 0$. Hence, $\|y_t - y^*(x_t)\| \leq \|y_t - y^*(x_{t-1})\| + \|y^*(x_t) - y^*(x_{t-1})\| \overset{t}{\to} 0$ using continuity of $y^*$, and $\|y_{t+1} - y_t\| \leq \|y_{t+1} - y^*(x_t)\| + \|y_t - y^*(x_t)\| \overset{t}{\to} 0$. $\qquad \square$

## C  PROOF OF THEOREM 1

**Theorem 1** (Global convergence). *Under the same conditions as those of Proposition 2, the Cubic-GDA has the following global convergence properties.*

1. *The function value sequence $\{\Phi(x_t)\}_t$ converges to a finite limit $H^* > -\infty$;*

2. *The generated sequences $\{x_t\}_t, \{y_t\}_t$ are bounded and have a compact sets of limit points.*

3. *Every limit point $x^*$ of $\{x_t\}_t$ is a second-order stationary point of $\Phi$, i.e., $\nabla\Phi(x^*) = \mathbf{0}$, $\nabla^2\Phi(x^*) \succeq \mathbf{0}$, and satisfies $\Phi(x^*) = H^*$.*

*Proof.* We first prove item 1. We have shown in Proposition 2 that $\{H_t\}_t$ is monotonically decreasing. Also, Assumption 1 says that $H_t \geq \Phi(x_t) \geq \inf_{x \in \mathbb{R}^m} \Phi(x) > -\infty$. Therefore, we conclude that $\{H_t\}_t$ converges to a finite limit $H^* > -\infty$. Since Proposition 2 proves that $\|x_t - x_{t-1}\|, \|y_{t+1} - y^*(x_t)\| \overset{t}{\to} 0$, we further conclude that $\Phi(x_t) \overset{t}{\to} H^*$, which proves item 1.

Next, we prove item 2. Note that $\Phi$ has compact sub-level sets. Moreover, for all $t$, we have

$$\Phi(x_t) \leq H_t \leq H_0 < +\infty. \tag{26}$$

Hence, the sequence $\{x_t\}_t$ is bounded and thus has a compact set of limit points. Since $y^*$ is Lipschitz continuous by Proposition 1, $\{\|y^*(x_t)\|\}_t$ is also bounded. Consequently, as $\|y_t - y^*(x_t)\| \to 0$, we conclude that $\{y_t\}_t$ is also bounded and hence has a compact set of limit points. This proves item 2.

Next, we will prove item 3. Suppose that $x_{t_k} \overset{k}{\to} x^*$ along a certain sub-sequence $\{x_{t_k}\}_k$. We have that $y_{t_k+1} \overset{t}{\to} y^*(x^*)$ based on Proposition 2 and the continuity of $y^*$. In addition, Proposition 1 implies that $\Phi, \nabla\Phi, \nabla^2\Phi$ and $G$ are continuous. Therefore, for every limit point $x^*$,

$$\Phi(x^*) = \lim_{t_k \to \infty} \Phi(x_{t_k}) = H^*.$$

$$\|\nabla\Phi(x^*)\| = \lim_{t_k\to\infty} \|\nabla\Phi(x_{t_k})\| = \lim_{t_k\to\infty} \|\nabla f(x_{t_k}, y^*(x_{t_k}))\| = \lim_{t_k\to\infty} \|\nabla f(x_{t_k}, y_{t_k+1})\|$$

$$\overset{(i)}{=} \lim_{t_k\to\infty} \left\| G(x_{t_k}, y_{t_k+1})(x_{t_k+1} - x_{t_k}) + \frac{1}{2\eta_x}\|x_{t_k+1} - x_{t_k}\|(x_{t_k+1} - x_{t_k}) \right\| = 0$$

$$\lambda_{\min}\big[\nabla^2\Phi(x^*)\big] = \lim_{t_k\to\infty} \lambda_{\min}\big[\nabla^2\Phi(x_{t_k})\big] = \lim_{t_k\to\infty} \lambda_{\min}\big[G(x_{t_k}, y^*(x_{t_k}))\big]$$

$$= \lim_{t_k\to\infty} \lambda_{\min}\big[G(x_{t_k}, y_{t_k+1})\big] \overset{(ii)}{\geq} -\frac{1}{2\eta_x} \lim_{t_k\to\infty} \|x_{t_k+1} - x_{t_k}\| = 0$$

where (i) uses eq. (22), and (ii) uses eq. (21). □

## D  PROOF OF THEOREM 2

**Theorem 2** (Global convergence rate). *Under the same conditions as those of Proposition 2, the Cubic-GDA converges at the following rate for all $T \geq \frac{H_0 - \inf_{x\in\mathbb{R}^m}\Phi(x)}{L_2\kappa^3/3}$.*

$$\min_{0\leq t\leq T-1} \mu(x_t) \leq \left(\frac{H_0 - \inf_{x\in\mathbb{R}^m}\Phi(x)}{TL_2\kappa^3/3}\right)^{1/3}.$$

Before proving Theorem 2, we first prove the following auxiliary lemma.

**Lemma 2.** *For any symmetric matrices $A, B \in \mathbb{R}^{n\times n}$, we have $|\lambda_{\min}(A) - \lambda_{\min}(B)| \leq \|A - B\|$.*

*Proof of Lemma 2.*

$$|\lambda_{\min}(A) - \lambda_{\min}(B)| \leq \Big| \min_{u:\|u\|=1} u^\top A u - \min_{u:\|u\|=1} u^\top B u \Big|$$

$$\leq \max_{u:\|u\|=1} \big| u^\top A u - u^\top B u \big|$$

$$\leq \max_{u:\|u\|=1} \big(\|u\|\|A - B\|\|u\|\big) = \|A - B\|.$$

□

*Proof of Theorem 2.* Note that (25) implies that

$$\min_{2\leq t\leq T-1} \big(\|x_{t+1} - x_t\|^3 + \|x_t - x_{t-1}\|^3\| + \|x_{t-1} - x_{t-2}\|^3\big) \leq \frac{H_0 - \inf_{x\in\mathbb{R}^m}\Phi(x)}{TL_2\kappa^3/3},$$

which further implies that there exists $1 \leq t' \leq T-1$ such that

$$\max\big(\|x_{t'+1} - x_{t'}\|, \|x_{t'} - x_{t'-1}\|, \|x_{t'-1} - x_{t'-2}\|\big) \leq \left(\frac{H_0 - \inf_{x\in\mathbb{R}^m}\Phi(x)}{TL_2\kappa^3/3}\right)^{1/3}. \quad (27)$$

On the other hand, equation (2.2) of Nesterov and Polyak (2006) implies that,

$$\|\nabla\Phi(x_{t'}) - \nabla\Phi(x_{t'-1}) - \nabla^2\Phi(x_{t'-1})(x_{t'} - x_{t'-1})\| \leq \frac{L_\Phi}{2}\|x_{t'} - x_{t'-1}\|^2.$$

Since $\nabla\Phi(x) = \nabla_1 f(x, y^*(x))$, $\nabla^2\Phi(x) = G(x, y^*(x))$, the above inequality implies that

$$\|\nabla\Phi(x_{t'})\|$$

$$\leq \|\nabla_1 f(x_{t'-1}, y^*(x_{t'-1})) + G(x_{t'-1}, y^*(x_{t'-1}))(x_{t'} - x_{t'-1})\| + \frac{L_\Phi}{2}\|x_{t'} - x_{t'-1}\|^2$$

$$\leq \|\nabla_1 f(x_{t'-1}, y_{t'}) + G(x_{t'-1}, y_{t'})(x_{t'} - x_{t'-1})\| + \|\nabla_1 f_1(x_{t'-1}, y^*(x_{t'-1})) - \nabla_1 f(x_{t'-1}, y_{t'})\|$$

$$+ \|\big(G(x_{t'-1}, y^*(x_{t'-1})) - G(x_{t'-1}, y_{t'})\big)(x_{t'} - x_{t'-1})\| + \frac{L_\Phi}{2}\|x_{t'} - x_{t'-1}\|^2$$

$$\overset{(i)}{\leq} \frac{1}{2\eta_x}\|x_{t'} - x_{t'-1}\|^2 + L_1\|y_{t'} - y^*(x_{t'-1})\| + L_G\|y_{t'} - y^*(x_{t'-1})\|\|x_{t'} - x_{t'-1}\|$$

$$+ \frac{L_\Phi}{2}\|x_{t'} - x_{t'-1}\|^2$$

$$\overset{(ii)}{\leq} \Big(\frac{1}{2\eta_x} + 4L_2\kappa^3\Big)\|x_{t'} - x_{t'-1}\|^2 + L_2\|x_{t'-1} - x_{t'-2}\|^2 + \frac{4L_2^2\kappa^2}{L_1}\|x_{t'} - x_{t'-1}\|^2\|x_{t'+1} - x_{t'}\|$$

$$\overset{(iii)}{\leq} \Big(\frac{1}{2\eta_x} + 5L_2\kappa^3\Big)\Big(\frac{H_0 - \inf_{x\in\mathbb{R}^m}\Phi(x)}{TL_2\kappa^3/3}\Big)^{2/3} + \frac{4L_2^2\kappa^2}{L_1}\Big(\frac{H_0 - \inf_{x\in\mathbb{R}^m}\Phi(x)}{TL_2\kappa^3/3}\Big)$$

$$\overset{(iv)}{\leq} \Big(\frac{1}{2\eta_x} + 5L_2\kappa^3 + \frac{4L_2^2\kappa^2}{L_1}\Big)\Big(\frac{H_0 - \inf_{x\in\mathbb{R}^m}\Phi(x)}{TL_2\kappa^3/3}\Big)^{2/3}, \tag{28}$$

where (i) uses eq. (22), item 1 of Assumption 1 and item 3 of Proposition 1, (ii) uses eq. (19) and $L_G = L_2(1+\kappa)^2 \leq 4L_2\kappa^2$, $L_\Phi = L_2(1+\kappa)^3 \leq 8L_2\kappa^3$, (iii) uses eq. (27), and (iv) uses $T \geq \frac{H_0 - \inf_{x\in\mathbb{R}^m}\Phi(x)}{L_2\kappa^3/3}$. Also, note that

$$-\lambda_{\min}\big(\nabla^2\Phi(x_{t'})\big)$$

$$\overset{(i)}{\leq} -\lambda_{\min}\big(G(x_{t'}, y_{t'+1})\big) + \|G(x_{t'}, y^*(x_{t'})) - G(x_{t'}, y_{t'+1})\|$$

$$\overset{(ii)}{\leq} \frac{1}{2\eta_x}\|x_{t'+1} - x_{t'}\| + L_G\|y_{t'+1} - y^*(x_{t'})\|$$

$$\overset{(iii)}{\leq} \frac{1}{2\eta_x}\|x_{t'+1} - x_{t'}\| + \frac{L_G L_2}{L_1}\|x_{t'} - x_{t'-1}\|^2$$

$$\overset{(iv)}{\leq} \frac{1}{2\eta_x}\Big(\frac{H_0 - \inf_{x\in\mathbb{R}^m}\Phi(x)}{TL_2\kappa^3/3}\Big)^{1/3} + \frac{L_G L_2}{L_1}\Big(\frac{H_0 - \inf_{x\in\mathbb{R}^m}\Phi(x)}{TL_2\kappa^3/3}\Big)^{2/3}$$

$$\overset{(v)}{\leq} \Big(\frac{1}{2\eta_x} + \frac{4L_2^2\kappa^2}{L_1}\Big)\Big(\frac{H_0 - \inf_{x\in\mathbb{R}^m}\Phi(x)}{TL_2\kappa^3/3}\Big)^{1/3}, \tag{29}$$

where (i) uses Lemma 2, (ii) uses eq. (21) and item 3 of Proposition 1, (iii) uses eq. (19), (iv) uses eq. (27), and (v) uses $T \geq \frac{H_0 - \inf_{x\in\mathbb{R}^m}\Phi(x)}{L_2\kappa^3/3}$ and $L_G = L_2(1+\kappa)^2 \leq 4L_2\kappa^2$. Equations (28) & (29) imply Theorem 2. □

## E   PROOF OF THEOREM 3

**Theorem 3.** *Let Assumption 1 hold and assume that the potential function $H$ satisfies the local Łojasiewicz gradient geometry. Choose the hyperparameters $N_t$, $\eta_x$, $\eta_y$ in the same way as Proposition 2. Then, the sequences $\{(x_t, y_t)\}_t$ generated by Cubic-GDA have a unique limit point, which is a second-order stationary point of $\Phi$.*

*Proof.* We first derive a bound on $\|\nabla H(x, x', y)\| := \sqrt{\sum_{k=1}^3 \|\nabla_k H(x, x', y)\|^2}$ as follows.

$$\|\nabla H(x_t, x_{t-1}, y_{t+1})\|$$
$$\leq \|\nabla_1 H(x_t, x_{t-1}, y_{t+1})\| + \|\nabla_2 H(x_t, x_{t-1}, y_{t+1})\| + \|\nabla_3 H(x_t, x_{t-1}, y_{t+1})\|$$
$$= \Big\|\nabla\Phi(x_t) + 3L_2\kappa^3\|x_t - x_{t-1}\|(x_t - x_{t-1}) + 12L_2\|y_{t+1} - y^*(x_t)\|\nabla y^*(x_t)^\top\big[y^*(x_t) - y_{t+1}\big]\Big\|$$
$$+ \Big\|3L_2\kappa^3\|x_t - x_{t-1}\|(x_t - x_{t-1})\Big\| + \Big\|12L_2\|y_{t+1} - y^*(x_t)\|\big[y_{t+1} - y^*(x_t)\big]\Big\|$$

$$\overset{(i)}{\leq} \|\nabla\Phi(x_{t-1}) + \nabla^2\Phi(x_{t-1})(x_t - x_{t-1})\| + \Big(\frac{L_\Phi}{2} + 6L_2\kappa^3\Big)\|x_t - x_{t-1}\|^2$$
$$+ 12L_2\|y_{t+1} - y^*(x_t)\|^2\Big(1 + \|\nabla_{22}f(x, y^*(x))^{-1}\|\|\nabla_{21}f(x, y^*(x))\|\Big)$$

$$\overset{(ii)}{\leq} \|\nabla\Phi(x_{t-1}) + G(x_{t-1}, y_t)(x_t - x_{t-1})\| + \|G(x_{t-1}, y^*(x_{t-1})) - G(x_{t-1}, y_t)\|\|x_t - x_{t-1}\|$$
$$+ \Big(\frac{L_\Phi}{2} + 6L_2\kappa^3\Big)\|x_t - x_{t-1}\|^2 + 12L_2(1 + \mu^{-1}L_1)\|y_{t+1} - y^*(x_t)\|^2$$

$$\overset{(iii)}{\leq} \Big\|\nabla_1 f(x_{t-1}, y^*(x_{t-1})) - \nabla_1 f(x_{t-1}, y_t) - \frac{1}{2\eta_x}\|x_t - x_{t-1}\|(x_t - x_{t-1})\Big\|$$

$$+ L_G\|y_t - y^*(x_{t-1})\|\|x_t - x_{t-1}\| + \left(\frac{L_\Phi}{2} + 6L_2\kappa^3\right)\|x_t - x_{t-1}\|^2 + 24L_2\kappa\|y_{t+1} - y^*(x_t)\|^2$$

$$\overset{(iv)}{\leq} L_1\|y_t - y^*(x_{t-1})\| + \left(\frac{L_\Phi}{2} + 6L_2\kappa^3 + \frac{1}{2\eta_x}\right)\|x_t - x_{t-1}\|^2$$

$$+ \frac{L_G L_2}{L_1}\|x_{t-1} - x_{t-2}\|^2\|x_t - x_{t-1}\| + \frac{24L_2^3\kappa}{L_1^2}\|x_t - x_{t-1}\|^4$$

$$\overset{(v)}{\leq} L_2\|x_{t-1} - x_{t-2}\|^2 + \left(10L_2\kappa^3 + \frac{1}{2\eta_x}\right)\|x_t - x_{t-1}\|^2$$

$$+ \frac{4L_2^2\kappa^2}{L_1}\|x_{t-1} - x_{t-2}\|^2\|x_t - x_{t-1}\| + \frac{24L_2^3\kappa}{L_1^2}\|x_t - x_{t-1}\|^4 \tag{30}$$

where (i) uses eq. (2.2) of Nesterov and Polyak (2006), the fact that $\nabla^2\Phi$ is $L_\Phi$-Lipschitz, and eq. (15), (ii) uses Lemma 1 and $\nabla^2\Phi(x) = G(x, y^*(x))$ from item 4 of Proposition 1, (iii) uses eq. (22), $\nabla\Phi(x) = \nabla_1 f(x, y^*(x))$ from item 2 of Proposition 1 and the inequality that $1 + \kappa \leq 2\kappa$, (iv) and (v) use eq. (19), and (v) uses $L_G = L_2(1+\kappa)^2 \leq 4L_2\kappa^2$, $L_\Phi = L_2(1+\kappa)^3 \leq 8L_2\kappa^3$.

Next, we prove the convergence of the sequence $\{x_t\}_t$ under the assumption that $H(x, x', y)$ satisfies the Łojasiewicz gradient geometry. Recall that we have shown in the proof of Theorem 1 that: 1) $\{H_t\}_t$ decreases monotonically to the finite limit $H^*$; 2) for any limit point $x^*, y^*$ of $\{x_t\}_t, \{y_t\}_t$, $\Phi(x^*) = H^*$. Hence, the Łojasiewicz gradient inequality (see Definition 1) holds after sufficiently large number of iterations, i.e., there exists $t_1 \in \mathbb{N}^+$ such that for all $t \geq t_1$, $H(x_t, x_{t-1}, y_{t+1}) - H^* \leq c\|\nabla H(x_t, x_{t-1}, y_{t+1})\|^\theta$. Equivalently,

$$\varphi'(H_t - H^*)\|\nabla H(x_t, x_{t-1}, y_{t+1})\| \geq 1.$$

where we define the concave function that $\varphi(s) := \frac{c^{1/\theta}}{1-1/\theta}s^{1-1/\theta}(\theta > 1, s > 0)$.

In addition, since $\|x_t - x_{t-1}\| \overset{t}{\to} 0$ (Proposition 2), there exists $t_2 \in \mathbb{N}^+$ such that for all $t \geq t_2$, $\|x_t - x_{t-1}\| \leq 1$. Hence, rearranging the above inequality and utilizing eq. (30), we obtain that for all $t \geq t_0 := \max(t_1, t_2)$,

$$\varphi'(H_t - H^*)$$
$$\geq \|\nabla H(x_t, x_{t-1}, y_{t+1})\|^{-1}$$
$$\geq \left[\left(\frac{4L_2^2\kappa^2}{L_1} + L_2\right)\|x_{t-1} - x_{t-2}\|^2 + \left(10L_2\kappa^3 + \frac{24L_2^3\kappa}{L_1^2} + \frac{1}{2\eta_x}\right)\|x_t - x_{t-1}\|^2\right]^{-1} \tag{31}$$

By concavity of the function $\varphi(s) := \frac{c^{1/\theta}}{1-1/\theta}s^{1-1/\theta}(\theta > 1, s > 0)$, we know that

$$\varphi(H_t - H^*) - \varphi(H_{t+1} - H^*)$$
$$\geq \varphi'(H_t - H^*)(H_t - H_{t+1})$$
$$\overset{(i)}{\geq} \frac{L_2\kappa^3\|x_{t+1} - x_t\|^3}{\left(\frac{4L_2^2\kappa^2}{L_1} + L_2\right)\|x_{t-1} - x_{t-2}\|^2 + \left(10L_2\kappa^3 + \frac{24L_2^3\kappa}{L_1^2} + \frac{1}{2\eta_x}\right)\|x_t - x_{t-1}\|^2}$$
$$\overset{(ii)}{\geq} \frac{L_2\kappa^3\|x_{t+1} - x_t\|^3}{\left(\sqrt{\frac{4L_2^2\kappa^2}{L_1} + L_2}\|x_{t-1} - x_{t-2}\| + \sqrt{10L_2\kappa^3 + \frac{24L_2^3\kappa}{L_1^2} + \frac{1}{2\eta_x}}\|x_t - x_{t-1}\|\right)^2}, \tag{32}$$

where (i) uses Proposition 2 and eq. (31), (ii) uses the inequality that $a^2 + b^2 \leq (a+b)^2$ for any $a, b \geq 0$.

Rearranging the above inequality yields that

$$L_2\kappa^3\|x_{t+1} - x_t\|^3$$
$$\leq [\varphi(H_t - H^*) - \varphi(H_{t+1} - H^*)]$$
$$\left(\sqrt{\frac{4L_2^2\kappa^2}{L_1} + L_2}\|x_{t-1} - x_{t-2}\| + \sqrt{10L_2\kappa^3 + \frac{24L_2^3\kappa}{L_1^2} + \frac{1}{2\eta_x}}\|x_t - x_{t-1}\|\right)^2$$

$$\leq \frac{1}{27}\Bigg( C^2[\varphi(H_t - H^*) - \varphi(H_{t+1} - H^*)] + \frac{2}{C}\sqrt{\frac{4L_2^2\kappa^2}{L_1} + L_2}\|x_{t-1} - x_{t-2}\|$$

$$+ \frac{2}{C}\sqrt{10L_2\kappa^3 + \frac{24L_2^3\kappa}{L_1^2} + \frac{1}{2\eta_x}}\|x_t - x_{t-1}\|\Bigg)^3 \tag{33}$$

where the final step uses the AM-GM inequality that $ab^2 = \left[\sqrt[3]{(C^2a)(b/C)(b/C)}\right]^3 \leq \frac{1}{27}(C^2a + \frac{2b}{C})^3$ for any $a, b \geq 0$ and $C > 0$ (the value of $C$ will be assigned later). Taking cubic root of both sides of the above inequality and telescoping it over $t = t_0, \ldots, T - 1$, we obtain that

$$\kappa\sqrt[3]{L_2}\sum_{t=t_0}^{T-1}\|x_{t+1} - x_t\|$$

$$\leq \frac{C^2}{3}[\varphi(H_{t_0} - H^*) - \varphi(H_T - H^*)] + \frac{2}{3C}\sqrt{\frac{4L_2^2\kappa^2}{L_1} + L_2}\sum_{t=t_0}^{T-1}\|x_{t-1} - x_{t-2}\|$$

$$+ \frac{2}{3C}\sqrt{10L_2\kappa^3 + \frac{24L_2^3\kappa}{L_1^2} + \frac{1}{2\eta_x}}\sum_{t=t_0}^{T-1}\|x_t - x_{t-1}\|$$

$$\leq \frac{C^2}{3}\varphi(H_{t_0} - H^*) + \frac{2}{3C}\sqrt{\frac{4L_2^2\kappa^2}{L_1} + L_2}\sum_{t=t_0}^{T-1}\|x_{t-1} - x_{t-2}\|$$

$$+ \frac{2}{3C}\sqrt{10L_2\kappa^3 + \frac{24L_2^3\kappa}{L_1^2} + \frac{1}{2\eta_x}}\sum_{t=t_0}^{T-1}\|x_t - x_{t-1}\|$$

where the final step uses the facts that $H_t - H^* \geq 0$ and that $\varphi(s)$ is monotonically increasing. Since the value of $C > 0$ is arbitrary, we can select large enough $C$ such that $\frac{2}{3C}\sqrt{\frac{4L_2^2\kappa^2}{L_1} + L_2}, \frac{2}{3C}\sqrt{10L_2\kappa^3 + \frac{24L_2^3\kappa}{L_1^2} + \frac{1}{2\eta_x}} < \frac{\kappa\sqrt[3]{L_2}}{3}$. Hence, the inequality above further implies that

$$\frac{\kappa\sqrt[3]{L_2}}{3}\sum_{t=t_0}^{T-1}\|x_{t+1} - x_t\| \leq \frac{C^2}{3}\varphi(H_{t_0} - H^*)$$

$$+ \frac{\kappa\sqrt[3]{L_2}}{3}\big[\|x_{t_0-1} - x_{t_0-2}\| + 2\|x_{t_0} - x_{t_0-1}\|\big] < +\infty. \tag{34}$$

Letting $T \to \infty$ concludes that

$$\sum_{t=1}^{\infty}\|x_{t+1} - x_t\| < +\infty.$$

Moreover, this implies that $\{x_t\}_t$ is a Cauchy sequence and therefore converges to a certain limit, i.e., $x_t \xrightarrow{t} x^*$. We have shown in Theorem 1 that any such limit point must be a second-order critical point of $\Phi$. Hence, we conclude that $\{x_t\}_t$ converges to a certain second-order critical point $x^*$ of $\Phi(x)$. Also, note that $\|y^*(x_t) - y_t\| \xrightarrow{t} 0$, $x_t \xrightarrow{t} x^*$ and $y^*$ is a Lipschitz mapping, so we conclude that $\{y_t\}_t$ converges to $y^*(x^*)$. Finally, the item 3 of Theorem 1 implies that $x^*$ is a second-order critical point of $\Phi(x)$. $\qquad\square$

# F    PROOF OF THEOREM 4

**Theorem 4** (Funtion value convergence rate). *Under the same conditions as those of Theorem 3, the sequence of potential function $\{H_t\}_t$ converges to the limit $H^*$ at the following rates.*

*1. If the geometry parameter $\theta \in (\frac{3}{2}, \infty)$, then $H_t \downarrow H^*$ super-linearly as*

$$H_t - H^* \leq \mathcal{O}\Big(\exp\Big(-\Big(\frac{2\theta}{3}\Big)^{\frac{t-t_0}{2}}\Big)\Big), \quad \forall t \geq t_0; \tag{6}$$

2. *If the geometry parameter $\theta = \frac{3}{2}$, then $H_t \downarrow H^*$ linearly as*

$$H_t - H^* \le (1 + C_0^{3/2})^{-\frac{t-t_0}{2}}, \quad \forall t \ge t_0; \tag{7}$$

3. *If the geometry parameter $\theta \in (1, \frac{3}{2})$, then $H_t \downarrow H^*$ sub-linearly as*

$$H_t - H^* \le \mathcal{O}\Big((t - t_0)^{-\frac{2\theta}{3-2\theta}}\Big), \quad \forall t \ge t_0. \tag{8}$$

*Proof.* Equation (31) implies that there exists $t_0 \in \mathbb{N}^+$ such that for any $t \ge t_0$,

$$
\begin{aligned}
\varphi'(H_t - H^*)^{-1} &= c^{-1/\theta}(H_t - H^*)^{1/\theta} \\
&\le \Big(\frac{4L_2^2\kappa^2}{L_1} + L_2\Big)\|x_{t-1} - x_{t-2}\|^2 + \Big(10L_2\kappa^3 + \frac{24L_2^3\kappa}{L_1^2} + \frac{1}{2\eta_x}\Big)\|x_t - x_{t-1}\|^2 \\
&\overset{(i)}{\le} \Big(\frac{4L_2^2}{L_1} + L_2\kappa^{-2}\Big)\Big(\frac{H_{t-2} - H_{t-1}}{L_2}\Big)^{2/3} + \Big(10L_2\kappa + \frac{24L_2^3}{L_1^2\kappa} + \frac{1}{2\eta_x\kappa^2}\Big)\Big(\frac{H_{t-1} - H_t}{L_2}\Big)^{2/3}
\end{aligned}
\tag{35}
$$

where (i) uses proposition 2.

Defining $d_t := H_t - H^*$, $C_1 := c^{1/\theta}L_2^{-2/3}\Big(\frac{4L_2^2}{L_1} + L_2\kappa^{-2}\Big)$, $C_2 := c^{1/\theta}L_2^{-2/3}\Big(10L_2\kappa + \frac{24L_2^3}{L_1^2\kappa} + \frac{1}{2\eta_x\kappa^2}\Big)$, and $C_0 = \sqrt[3]{2}c^{1/\theta}L_2^{-2/3}\Big(10L_2\kappa + \frac{24L_2^3}{L_1^2\kappa} + \frac{4L_2^2}{L_1} + \frac{1}{2\eta_x\kappa^2}\Big)$, the above inequality further becomes

$$
\begin{aligned}
d_t^{1/\theta} &\le C_1\big(d_{t-2} - d_{t-1}\big)^{2/3} + C_2\big(d_{t-1} - d_t\big)^{2/3} \tag{36} \\
&\le 2\max(C_1, C_2)\Big[\frac{1}{2}\big(d_{t-2} - d_{t-1}\big)^{2/3} + \frac{1}{2}\big(d_{t-1} - d_t\big)^{2/3}\Big] \\
&\overset{(i)}{\le} 2\max(C_1, C_2)\Big[\frac{1}{2}(d_{t-2} - d_t)\Big]^{2/3} \overset{(ii)}{\le} C_0(d_{t-2} - d_t)^{2/3} \tag{37}
\end{aligned}
$$

where (i) applies Jensen's inequality to the concave function $\xi(s) = s^{2/3}$, and (ii) uses the inequality that $2^{1/3}\max(C_1, C_2) \le C_0$.

Next, we prove the convergence rates case by case.

(Case 1) If $\theta \in (\frac{3}{2}, +\infty)$, since $d_t \ge 0$, eq. (36) implies that for $t \ge t_0$,

$$d_t \le C_0^\theta d_{t-2}^{2\theta/3},$$

which is equivalent to that

$$C_0^{\frac{3\theta}{2\theta-3}} d_t \le \big[C_0^{\frac{3\theta}{2\theta-3}} d_{t-2}\big]^{2\theta/3} \tag{38}$$

Since $d_t \downarrow 0$, $C_0^{\frac{3\theta}{2\theta-3}} d_{t_0} \le e^{-1}$ for sufficiently large $t_0 \in \mathbb{N}^+$. Hence, eq. (38) implies that for any $k \in \mathbb{N}^+$

$$C_0^{\frac{3\theta}{2\theta-3}} d_{2k+t_0} \le \big[C_0^{\frac{3\theta}{2\theta-3}} d_{t_0}\big]^{[(2\theta/3)^k]} \le \exp\Big[-\Big(\frac{2\theta}{3}\Big)^k\Big].$$

Hence,

$$d_{2k+t_0+1} \le d_{2k+t_0} \le C_0^{-\frac{3\theta}{2\theta-3}} \exp\Big[-\Big(\frac{2\theta}{3}\Big)^k\Big]$$

Note that $\theta \in (\frac{3}{2}, +\infty)$ implies that $\frac{2\theta}{3} > 1$, and thus the inequality above implies that $H_t \downarrow H^*$ at the super-linear rate given by eq. (6).

(Case 2) If $\theta = \frac{3}{2}$, eq. (37) implies that for $t \ge t_0$,

$$d_t \le (1 + C_0^{3/2})^{-1} d_{t-2}. \tag{39}$$

Note that $d_{t_0} \leq 1$ for sufficiently large $t_0$. Therefore, $d_t \downarrow 0$ (i.e., $H(z_t) \downarrow H^*$) at the linear rate given by eq. (7).

(Case 3) If $\theta \in (1, \frac{3}{2})$, consider the following two subcases.

If $d_{t-2} \leq 2d_t$, denote $\psi(s) = \frac{2\theta}{3-2\theta} s^{1-\frac{3}{2\theta}}$, then for any $t \geq t_0$,

$$\psi(d_t) - \psi(d_{t-2}) = \int_{d_t}^{d_{t-2}} -\psi'(s)ds = \int_{d_t}^{d_{t-2}} s^{-\frac{3}{2\theta}} ds \overset{(i)}{\geq} d_{t-2}^{-\frac{3}{2\theta}} (d_{t-2} - d_t)$$

$$\overset{(ii)}{\geq} C_0^{-\frac{3}{2}} \left( \frac{d_t}{d_{t-2}} \right)^{\frac{3}{2\theta}} \overset{(iii)}{\geq} (2C_0)^{-\frac{3}{2}} \tag{40}$$

where (i) uses $d_t \leq d_{t-2}$, and $-\frac{3}{2}(1-\theta) < 0$, (ii) uses the following inequality implied by eq. (37), and (iii) uses $\frac{d_t}{d_{t-2}} \geq \frac{1}{2}$ and $\frac{3}{2\theta} \in (1, \frac{3}{2})$.

If $d_{t-2} > 2d_t$, then for any $t \geq t_0$,

$$\psi(d_t) - \psi(d_{t-2}) = \frac{2\theta}{3-2\theta} (d_t^{1-\frac{3}{2\theta}} - d_{t-2}^{1-\frac{3}{2\theta}}) \geq \frac{2\theta}{3-2\theta} \left[ d_t^{1-\frac{3}{2\theta}} - (2d_t)^{1-\frac{3}{2\theta}} \right]$$

$$\geq \frac{2\theta \left( 1 - 2^{1-\frac{3}{2\theta}} \right)}{3-2\theta} d_t^{1-\frac{3}{2\theta}} \geq (2C_0)^{-\frac{3}{2}}. \tag{41}$$

where we use $1 - \frac{3}{2\theta} \in \left( -\frac{1}{2}, 0 \right)$, $\frac{2\theta}{3-2\theta} > 0$ and $d_t \leq d_{t_0} \leq (2C_0)^{\frac{3\theta}{3-2\theta}} \left[ \frac{2\theta \left( 1-2^{1-\frac{3}{2\theta}} \right)}{3-2\theta} \right]^{\frac{2\theta}{3-2\theta}}$ for sufficiently large $t_0 \in \mathbb{N}^+$.

Since at least one of eqs. (40) & (41) holds, we have $\psi(d_t) - \psi(d_{t-2}) \geq (2C_0)^{-\frac{3}{2}}$. Hence,

$$\psi(d_{2k+t_0+1}) \geq \psi(d_{2k+t_0}) \geq \psi(d_{t_0}) + k(2C_0)^{-\frac{3}{2}} \geq k(2C_0)^{-\frac{3}{2}}; k \in \mathbb{N},$$

which implies that $\psi(d_t) \geq \frac{t-t_0}{2}(2C_0)^{-\frac{3}{2}}$ By substituing the definition of $\psi$, the inequality above implies that $H(z_t) \downarrow H^*$ in a sub-linear rate given by eq. (8). $\qquad \square$

## G    PROOF OF THEOREM 5

**Theorem 5** (Parameter convergence rate). *Under the same conditions as those of Theorem 3, the sequences $\{x_t, y_t\}_t$ generated by Cubic-GDA converge to their limits $x^*, y^*(x^*)$ respectively at the following rates.*

1. *If the geometry parameter $\theta \in (\frac{3}{2}, \infty)$, then $(x_t, y_t) \to (x^*, y^*(x^*))$ super-linearly as*

$$\max \left\{ \|x_t - x^*\|, \|y_t - y^*(x^*)\| \right\} \leq \mathcal{O}\left( \exp\left( -\frac{1}{3} \left( \frac{2\theta}{3} \right)^{\frac{t-t_0}{2}-1} \right) \right), \quad \forall t \geq t_0; \tag{9}$$

2. *If the geometry parameter $\theta = \frac{3}{2}$, then $(x_t, y_t) \to (x^*, y^*(x^*))$ linearly as*

$$\max \left\{ \|x_t - x^*\|, \|y_t - y^*(x^*)\| \right\} \leq \mathcal{O}\left( (1+C_0^{3/2})^{-\frac{t-t_0}{2}} \right), \quad \forall t \geq t_0; \tag{10}$$

3. *If the geometry parameter $\theta \in (1, \frac{3}{2})$, then $(x_t, y_t) \to (x^*, y^*(x^*))$ sub-linearly as*

$$\|x_t - x^*\| \leq \mathcal{O}\left( (t-t_0)^{-\frac{2(\theta-1)}{3-2\theta}} \right), \quad \|y_t - y^*(x_t)\| \leq \mathcal{O}\left( (t-t_0)^{-\frac{2\theta}{3(3-2\theta)}} \right), \quad \forall t \geq t_0. \tag{11}$$

*Proof.* Notice that eq. (34) still holds after increasing $t_0$, i.e., for $T \geq t \geq t_0$ and large enough $C$ such that $\frac{2}{3C}\sqrt{\frac{4L_2^2\kappa^2}{L_1} + L_2}, \frac{2}{3C}\sqrt{10L_2\kappa^3 + \frac{24L_2^3\kappa}{L_1^2} + \frac{1}{2\eta_x}} < \frac{\kappa\sqrt[3]{L_2}}{3}$, we have

$$\frac{\kappa\sqrt[3]{L_2}}{3} \sum_{s=t}^{T-1} \|x_{s+1} - x_s\| \leq \frac{C^2 c^{1/\theta}}{3(1-1/\theta)} (H_t - H^*)^{1-1/\theta}$$

$$+ \frac{\kappa\sqrt[3]{L_2}}{3} \left[ \|x_{t-1} - x_{t-2}\| + 2\|x_t - x_{t-1}\| \right] < +\infty. \tag{42}$$

Proposition 2 implies that

$$\|x_t - x_{t-1}\| \leq L_2^{-1/3}\kappa^{-1}(H_{t-1} - H_t)^{1/3} \overset{(i)}{\leq} L_2^{-1/3}\kappa^{-1}(H_{t-1} - H^*)^{1/3}, \tag{43}$$

where (i) uses $H_{t-1} \geq H_t \geq H^*$.

Therefore, for $t \geq t_0$, eqs. (42) & (43) imply that

$$\begin{aligned}
\|x_t - x^*\| &\leq \limsup_{T \to \infty} \sum_{s=t}^{T-1} \|x_{s+1} - x_s\| \\
&\leq \frac{C^2 c^{1/\theta}}{\kappa \sqrt[3]{L_2}(1 - 1/\theta)}(H_t - H^*)^{1-1/\theta} + \left[(H_{t-2} - H^*)^{1/3} + 2(H_{t-1} - H^*)^{1/3}\right]
\end{aligned} \tag{44}$$

Next, we discuss case by case.

(Case 1) If $\theta \in \left(\frac{3}{2}, +\infty\right)$, then

$$\begin{aligned}
\|x_t - x^*\| &\leq \mathcal{O}\left[(H_{t-2} - H^*)^{1/3} + (H_{t-1} - H^*)^{1/3} + (H_t - H^*)^{1/3}\right] \tag{45} \\
&\leq \mathcal{O}\left(\exp\left[-\frac{1}{3}\left(\frac{2\theta}{3}\right)^{(t-t_0)/2-1}\right]\right)
\end{aligned}$$

where the two $\leq$ use eqs. (44) & (6) respectively. Hence,

$$\begin{aligned}
\|y_t - y^*(x_t)\| &\leq \|y_t - y^*(x_{t-1})\| + \|y^*(x_{t-1}) - y^*(x_t)\| \\
&\overset{(i)}{\leq} \frac{L_2}{L_1}\|x_{t-1} - x_{t-2}\|^2 + \kappa\|x_t - x_{t-1}\| \tag{46} \\
&\overset{(ii)}{\leq} \mathcal{O}\left[(H_{t-2} - H^*)^{2/3} + (H_{t-1} - H^*)^{1/3}\right] \tag{47} \\
&\leq \mathcal{O}\left(\exp\left[-\frac{1}{3}\left(\frac{2\theta}{3}\right)^{(t-t_0-2)}\right] + \exp\left[-\frac{1}{3}\left(\frac{2\theta}{3}\right)^{(t-t_0-1)/2}\right]\right) \\
&\leq \mathcal{O}\left(\exp\left[-\frac{1}{3}\left(\frac{2\theta}{3}\right)^{(t-t_0-1)/2}\right]\right),
\end{aligned}$$

where (i) uses eq. (19) and item 1 of Proposition 1, (ii) uses eq. (43) and (iii) uses eq. (6).

(Case 2) If $\theta = \frac{3}{2}$, then the proof is similar to that of Case 2, and eqs. (45) & (47) still hold. The only difference is to use the convergence rate of $H_t - H^*$ given by eq. (7) instead of eq. (6).

(Case 3) If $\theta \in (1, \frac{3}{2})$, then similar to the proof of Case 2, we obtain from eq. (44) that

$$\begin{aligned}
\|x_t - x^*\| &\leq \mathcal{O}\left[(H_{t-2} - H^*)^{1-1/\theta} + (H_{t-1} - H^*)^{1-1/\theta} + (H_t - H^*)^{1-1/\theta}\right] \\
&\leq \mathcal{O}\left((t - t_0)^{-\frac{2(\theta-1)}{3-2\theta}}\right),
\end{aligned}$$

where the two $\leq$ use eqs. (43) & (6) respectively. Then, as eq. (47) still holds, we have

$$\|y_t - y^*(x_t)\| \leq \mathcal{O}\left((t - t_0)^{-\frac{2\theta}{3(3-2\theta)}}\right).$$

$\square$

