# OpenReview forum: "Escaping Saddle Points in Nonconvex Minimax Optimization via Cubic-Regularized Gradient Descent-Ascent"
_ICLR.cc/2022/Conference — ICLR 2022 Submitted_

### Official Review · Reviewer_c2gL · 2021-10-18

**Correctness:** 4
**Technical Novelty And Significance:** 3
**Empirical Novelty And Significance:** 3
**Recommendation:** 8
**Confidence:** 3

**Main Review:**

The paper combines gradient ascent-descent techniques with the second-order optimization approach of cubic regularization, leveraging on the preservation of Lipschitz properties of the solution of the max problem, while using only an approximated solution for the strongly concave max problem.
I found the approach interesting and sufficiently innovative.
The paper is well-written, the mathematical  writing and analysis are nice and straightforward, and their combination of techniques is interesting.
I read most of the proofs in the appendix, and to the best of my knowledge, are correct, although I did not read too carefully; nonetheless, the results are sensible and coherent compared to standard analysis and known results.

A few issues:

1. The structure of the paper, or more precisely the locations of the problem formulation and baseline assumption, is confusing for the expert reader. The problem is declared in 3 locations, together with the definition of second-order stationarity, but the first two statements are lacking in terms of mathematical formulation. This can be solved easily by moving Assumption 1 to the beginning of the paper, so all the statements made up until page 4 will make sense...
Additionally, in the same context, pages 1-4 indeed feel a bit repetitive, consider maybe revising, and then you will have some spaces for additional things, such as proofs sketch, etc.
2. The cubic regularization approach was first taken by: Griewank, A.: The Modification of Newton’s Method for Unconstrained Optimization by Bounding Cubic Terms. Technical Report NA/12 (1981).
Unfortunately, this is quite an unknown manuscript in the community that deserves the credit for being the first (to the best of my knowledge).
3. Another recent paper for second-order optimization that might be relevant and was not mentioned is: Hallak, N., and Teboulle, M., "Finding second-order stationary points in constrained minimization: A feasible direction approach." Journal of Optimization Theory and Applications 186.2 (2020): 480-503.
I think this paper also has a version for unconstrained problems.
4. There is a typo in the declaration of Theorem 4.


**Summary Of The Paper:**

The paper proposes a second-order cubic regularization type method for an unconstrained min-max problem in which the function is twice continuously differential, strongly concave w.r.t the max decision variable, with Lipschitz continuous first and second derivatives.
The proposed method comprises a nested gradient ascent process of the max variable, with a dynamic number of iterations that guarantee the desired properties to support the convergence resutls, and a cubic proximal update on the min variable.
The method is proved to converge to second-order stationary point, under standard assumptions.

**Summary Of The Review:**

Pros:
- New method combining elements from first and second order methods
- The method only approximates the solution of the max problem
- New convergence guarantees for the min-max problem
- Well-written mathematically and in general

Cons:
- Results are not surprising, and somewhat a direct implication, considering the techniques used and the current literature.

Overall: I recommend to accept the paper.

---

> ### Author Response · Authors · 2021-11-18
> **Response to Review**
>
> Thank you very much for reviewing our manuscript and providing valuable feed-back.  Below is a response to the review comments. We have submitted a revised version with all revisions marked in `red`.  Please let us know if further clarifications are needed.
>
> **Q1:** Consider removing repetitive contents and moving Assumption 1 to the beginning of the paper.
>
> **A:** Thank you for the great suggestion. In the revision, we have removed the repetitive problem definitions and contents. We also moved Assumption 1 to the beginning of Section 2. In the contribution section, we also mentioned the major conditions of Assumption 1.
>
> **Q2:** The cubic regularization approach was first taken by: Griewank, A.: The Modification of Newton’s Method for Unconstrained Optimization by Bounding Cubic Terms. Technical Report NA/12 (1981).
>
> **A:** Thank you very much for pointing out this reference. It is excited to know about this work. We have cited this work in the related work in our revision.
>
> **Q3:** The paper Hallak, N. and Teboulle, M. 2020 works on unconstrained second-order optimization.
>
> **A:** Thank you for pointing out this related work. We have cited and discussed this work in the related work in our revision.
>
> **Q4:** There is a typo in the declaration of Theorem 4.
>
> **A:** We did not notice any typo in Theorem 4. Can you kindly point it out? Thanks!

---

### Official Review · Reviewer_Xmar · 2021-10-29

**Correctness:** 4
**Technical Novelty And Significance:** 4
**Empirical Novelty And Significance:** 3
**Recommendation:** 6
**Confidence:** 5

**Main Review:**

This work develops a non-trivial extension of first-order GDA algorithm to second-order algorithms for escaping saddle points in nonconvex minimax optimization. Specifically, the Cubic-GDA algorithm uses gradient ascent to estimate the second-order information of the minimax objective function, and leverages the cubic regularization technique to efficiently escape the strict saddle points.

The paper provides a comprehensive convergence analysis of this algorithm. Including its global convergence properties and convergence rates in general nonconvex minimax optimization, as well as its convergence rates under a broad class of gradient dominant conditions. The analysis is based on a careful characterization of the Jacobian matrix of the minimax objective function and its Lipschitz continuity property (Prop. 1). The key to the analysis is characterization of a special potential function that is proven to be monotonically decreasing (Prop. 2). This is an elegant result that justifies the numerical stability and correctness of the algorithm.

Under a wide spectrum of gradient dominant conditions, it is proved that Cubic-GDA achieves an orderwise faster convergence rate than the standard GDA. This demonstrates the effectiveness of introducing cubic regularization into nonconvex minimax optimization. This may inspire future developments of second-order minimax optimization algorithms.


The proposed Cubic-GDA algorithm can be viewed as an inexact cubic-regularization algorithm. Can the existing analysis of inexact Cubic regularization be applied here?

The learning rate $\eta_x$ has a bad dependence of the problem condition number. This seems to be caused by the bounding techniques used in the proof. Is there a way to further reduce the dependence? Note that the GDA studied in (Lin 2020) only has a dependence of $\kappa^2$.

If we apply accelerated gradient descent to solve the strongly concave sub problem, then it may lead to a faster global convergence rate of the algorithm. I suggest the authors add more discussions on this point.





**Summary Of The Paper:**

This paper develops Cubic-GDA—the first GDA-type algorithm that can find second-order stationary points in nonconvex minimax optimization, whereas the existing GDA algorithms can only converge to first-order stationary points. The developed algorithm leverages the popular Cubic regularization technique to escape saddle points in nonconvex minimax optimization.

Under standard assumptions, the authors identify a monotonically decreasing potential function of the proposed Cubic-GDA algorithm, and leverage it to establish global convergence of Cubic-GDA to a second-order stationary point. Moreover, they analyze the convergence rate of Cubic-GDA in the full spectrum of gradient dominant-type geometries, and show that Cubic-GDA achieves an orderwise faster convergence rate than the standard GDA.


**Summary Of The Review:**

Overall, this paper is nicely written and contains solid theoretical contributions to nonconvex minimax optimization. Given the emerging and growing interests in minimax optimization in the general machine learning community, I recommend accepting this work.

-----------------------------
After reading the other reviews, it seems that the present version indeed consists of a number of issues and is not yet ready for publication. But overall, I feel the paper contributes to analysis and algorithm design for minimax optimization. I have updated my score.

---

> ### Author Response · Authors · 2021-11-18
> **Response to Review**
>
> Thank you very much for reviewing our manuscript and providing valuable feed-back.  Below is a response to the review comments. We have submitted a revised version with all revisions marked in `red`.  Please let us know if further clarifications are needed.
>
> **Q1:** Can the existing analysis of inexact Cubic regularization be applied here?
>
> **A:** Great question. While our analysis can be viewed as an inexact-type analysis of CR, it is different from the existing analysis of inexact CR in several perspectives. First, in the existing literature on inexact CR, the inexactness is often induced by stochastic subsampling when the objective function takes a finite-sum form over a large number of samples. As a comparison, our inexactness is induced by solving the strongly concave problem via gradient ascent. Second, conventional inexact CR usually controls the level of inexactness introduced in every cubic update below a pre-determined constant tolerance level $\epsilon$ in order to guarantee global convergence, but this cannot establish a proper decreasing potential function, which is the key to develop all the analysis of this work. As a comparison, our analysis adapt the level of inexactness in every iteration and establishes the potential function in our Proposition 2 that allows us to prove global convergence as well as local convergence of the algorithm.
>
> **Q2:** The learning rate has poor dependence on $\kappa$.
>
> **A:** Great question. We found that the dependence of our stepsize on the condition number can be substantially improved, from $\mathcal{O}(\kappa^{-7})$ to $\mathcal{O}(\kappa^{-3})$, by simply optimizing the
> coefficients introduced in several elementary AM-GM inequalities. Specifically, in the revision, we update the proof of the key Proposition 2 and optimize the use of AM-GM inequalities in eqs.(20, 23) to obtain the best dependence of $\eta_x$ on $\kappa$. We have also updated the other results accordingly. Please refer to our revised version for the details.
>
> **Q3:** How about using accelerated gradient descent to the strongly concave sub problem?
>
> **A:** Great question. Yes, accelerated gradient ascent can be directly applied here to further improve the convergence rate and complexity of Cubic-GDA. Specifically, with momentum acceleration, the convergence rate of the gradient ascent for solving the strongly concave function $f(x_t, \cdot)$ achieves a faster convergence rate, i.e., the rate in eq. (18) in the appendix can be improved to $||y_{t+1} - y^*(x_{t}) || \le   (1-\sqrt{\kappa}^{-1})^{N_t} ||y_{t} - y^*(x_{t})||$. With this accelerated convergence rate, we can follow the same analysis logic and establish a global convergence rate of Cubic-GDA with better dependence on the condition number $\kappa$.

---

### Official Review · Reviewer_ENLL · 2021-11-02

**Correctness:** 4
**Technical Novelty And Significance:** 2
**Empirical Novelty And Significance:** Not applicable
**Recommendation:** 3
**Confidence:** 5

**Main Review:**

Finding second-order stationary points of a nonconvex minimax problem is an interesting and important problem in the optimization area. The proposed algorithm seems intuitive and reasonable with clear motivation. However, the theoretical contributions of the paper are not strong and the empirical study is missing.

My first concern is about the global convergence rate. According to Proposition 2, $\eta_x$ should be $O(\kappa^9)$. Then, the global convergence rate in Theorem 2 shows that cubic-GDA seems to require $O(\kappa^{13.5}\epsilon^{-1.5})$ total iterations to achieve an $\epsilon$-first-order stationary point which satisfies $||\nabla\Phi(x_t)||\leq \epsilon$. Notice that the convergence rate of GDA is only $O(\kappa^2\epsilon^{-2})$ (Lin et. al., 2020). The order of $\kappa$ in the results of Cubic-GDA  is much worse than that of GDA.
Such result is unbearable for ill-conditioned problems (which is very common in real applications).

My second concern is about the way of solving the cubic regularization sub-problem. The paper claims that the cubic regularized sub-problem can be efficiently solved by gradient descent methods (Carmon and Duchi, 2016). However, gradient descent can only achieve a sublinear convergence rate.
Hence, I think it is important to discuss how the inexact minimizer affects the convergence behavior of Cubic-GDA.
The theoretical analysis of this paper is based on assuming the sub-problem can be solved exactly, which is not reasonable.

My last concern is this paper does not provide any empirical results. The previous two concerns show that the proposed algorithm may converge very slowly. Thus I doubt whether the proposed method can outperform GDA algorithms in real applications. I hope the authors perform some empirical studies to verify the effectiveness of the proposed algorithm. Additionally, it is preferred to conduct experiments to show Cubic-GDA can escape some unexpected suboptimal saddle points that GDA stuck.

Minors:

Do you have any ideas to generalize current results to nonconvex-concave (not strongly-concave) setting?

**Summary Of The Paper:**

This paper studies nonconvex minimax optimization problems and proposes a novel algorithm, called Cubic-GDA, which can escape the strict saddle points and find the second-order stationary points of the nonconvex minimax problem. The paper also provides the global convergence rate of the proposed algorithm and analyzes the convergence property under local nonconvex geometry.

**Summary Of The Review:**

Pros:
1. The studied problem is interesting and the proposed method is novel.

Cons:
1. Theoretical analysis is on the weak side.
2. No empirical results.

Overall, I recommend to reject this paper.

---

> ### Author Response · Authors · 2021-11-18
> **Response to Review**
>
> Thank you very much for reviewing our manuscript and providing valuable feed-back.  Below is a response to the review comments. We have submitted a revised version with all revisions marked in `red`.  Please let us know if further clarifications are needed.
>
> **Q1:** The stepsize and convergence rate has poor dependence on $\kappa$.
>
> **A:** Great question. We found that the dependence of our stepsize on the condition number can be substantially improved, from $\mathcal{O}(\kappa^{-7})$ to $\mathcal{O}(\kappa^{-3})$, by simply optimizing the
> coefficients introduced in several elementary AM-GM inequalities. Specifically, in the revision, we update the proof of the key Proposition 2 and optimize the use of AM-GM inequalities in eqs.(20, 23) to obtain the best dependence of $\eta_x$ on $\kappa$. We have also updated the other results accordingly. Please refer to our revised version for the details.
>
> **Q2:** Gradient descent in (Carmon and Duchi, 2016) can only solve the cubic sub-problem with a sublinear convergence rate. The theoretical analysis is based on assuming the sub-problem can be solved exactly, it is important to discuss how inexact minimizer affects the convergence behavior of Cubic-GDA.
>
> **A:** Thanks for the suggestion and comments. We first want to clarify that Gradient descent in (Carmon and Duchi, 2016) is proved to achieve a sub-linear convergence rate only when the target accuracy is large, and it actually achieves a much faster linear convergence rate when the target accuracy is small. Therefore, we think this solver is efficient to produce a high-accuracy solution.
>
> Regarding the reviewer's comment on inexact optimization, we want to point out that our analysis framework of Cubic-GDA can allow inexactness in formulating the cubic subproblem, by leveraging the large body of existing development on Inexact Cubic Regularization. Specifically, instead of solving the cubic subproblem with the exact gradient $\nabla_1 f(x_t, y_{t+1})$ and Jacobian $G(x_t,y_{t+1})$, it suffices to use their inexact versions $p_t$ and $P_t$ that satisfies the conditions
> $||p_t - \nabla_1 f(x_t, y_{t+1})|| \le O(||x_{t+1} - x_{t}||^2)$, $||P_t - G(x_t,y_{t+1})||\le O(||x_{t+1} - x_{t}||)$. Such type of inexactness conditions are widely used and studied in the existing literature on cubic regularization, e.g.,
>
> [1]'Adaptive cubic regularization methods for unconstrained optimization'.
>
> Under such inexact conditions, most of our proof of the key Proposition 2 remains unchanged, except that the coefficients of the term $||x_{t+1} - x_t||^3$ is slightly different. We have added a remark on inexactness after the Proposition 2 in the revision.
>
>
> **Q3:** Any ideas to generalize current results to nonconvex-concave setting?
>
> **A:** This is an interesting direction for future research. One potential way is to use smoothing techniques, i.e., make the concave part strongly concave by adding a small strongly concave regularizer, and study how the solution of the nonconvex-strongly-concave problem deviates from that of the original nonconvex-concave problem.

---

> > ### Comment · Reviewer_ENLL · 2021-11-29
> > **Thank you for your reply**
> >
> > After reading your response and other reviews, I have decided to keep my score. I'm disappointed that the authors did not reply to reviewers' concerns about the lack of empirical results. Also, the discussion about the inexact version of Cubic-GDA is somewhat rough. Though inexactness in formulating the cubic subproblem has been widely studied in nonconvex setting, I think it is not trivial to adopt it on the minimax problem. I suggest authors give more detailed analysis on the convergence property, such as the number of Hessian-vector products for finding an $\epsilon$-stationary point.

---

> > > ### Author Response · Authors · 2021-11-30
> > > **Thanks for the update**
> > >
> > > Thanks a lot for the update and we respect your decision. The focus of this work is to 1) propose the new Cubic-GDA algorithm for minimax optimization and 2) provide a comprehensive understanding of its dynamics as well as global/local convergence rates under various nonconvex geometries, which we think are valuable aspects. We agree that analyzing the computation complexity involving Hessian inexactness is another important aspect, but providing a comprehensive and quantitive characterization would require a separate study under a possibly very different setting (such as the stochastic setting).
> > >
> > > As a side note, the first paper (Nesterov, 2006) that studies CR in nonconvex optimization also focuses on analyzing the global/local convergence rates under different nonconvex geometries. It inspired numerous follow-up works that developed inexact versions of the CR algorithm and proposed computation efficient solvers.

---

### Official Review · Reviewer_W1TD · 2021-11-03

**Correctness:** 4
**Technical Novelty And Significance:** 2
**Empirical Novelty And Significance:** 1
**Recommendation:** 3
**Confidence:** 4

**Main Review:**

Major issues:

1. The key takeaway is that Cubic-GDA converges to a “second-order stationary point” of the envelope function. While it is well-received in nonconvex optimization literature, the authors could explain why such a point is of interest in the context of minimax optimization, i.e., how it is related to the solution of (P). For instance, it appears to have connection with the second-order necessary condition for a local minimax point introduced in (Jin et al. 2020, “What is local optimality in nonconvex-nonconcave minimax optimization?”).

2. The paper is not very novel in terms of algorithms development cause it basically runs the update of CRN method on the envelope function and reaches its second-order stationary point, which is known for CRN. The only technical challenge is addressing the accuracy required for solving the inner maximization problem. Indeed, this is not very challenging as in the strongly concave case the inner problems can be solved exponentially fast.

3. The comparisons with related work seem insufficient. I expect the authors to discuss more on current first-order methods for the nonconvex-strongly-concave minimax problems. Also, given that the proposed method is in essence an inexact version of CR on the envelope function, the authors may need to justify why existing results on inexact CR method are not directly applicable.

4. In Proposition 2, the choice of $\eta_x$ has a very bad dependence on the conditional number (on the order of \kappa^{-7}). This in turn leads to huge constants in most of the convergence results, which could potentially undo the benefit of using second-order methods if the required accuracy is moderate.

5. The original CR method in (Nesterov and Polyak, 2006) is also shown to behave very well in the local neighborhood of a strict saddle point or a strict local minimum. So I am wondering whether cubic-GDA can retain these desired features or not.

6. It will be helpful if the authors can include numerical experiments to support the theoretical findings. For instance, it will make a very good point if the authors can come up with a specific example where GDA converges to an undesired stationary point while cubic-GDA succeeds in finding a local minimax solution.

Minor issues:

1. At the bottom of page 1: “envelop” -> “envelope”.
2. The paper refers to y^*(x) as the minimizer, but it is in fact the maximizer.
3. The authors describe their method as a “GDA-type algorithm”, which is a bit misleading. First of all, there is no gradient descent step. Also, since the update of x requires second-order information, it seems more appropriate to address it as a Newton-type method.
4. Both “Jin et al. 2017a” and “Jin et al. 2017b” point to the same reference. Also, “Jin et al. 2017b” mentioned in the related work on CR seems to be the wrong reference.

**Summary Of The Paper:**


In this paper, the authors develop a second-order method based on the cubic regularization technique for solving nonconvex-strongly-concave minimax problems. They analyze its convergence rate under the standard smoothness assumptions as well as under the local Lojasiewicz gradient geometry. Notably, this is the first algorithm that provably converges to a second-order stationary point and hence escapes strict saddle points.

**Summary Of The Review:**

I think the paper addresses an interesting problem, but its algorithm development is not very novel. Moreover, the authors need to better explain why achieving such second-order stationary point should be of interest in the context of minimax optimization.

---

> ### Author Response · Authors · 2021-11-18
> **Response to Review**
>
> Thank you very much for reviewing our manuscript and providing valuable feedback. Below is a response to the major review comments. We have submitted a revised version with all revisions marked in `red`. Please let us know if further clarifications are needed.
>
> **Q1:** How does second-order staiontary point of the envelop function relate to the minimax optimization $(P)$? How does it relate to the second-order necessary condition in Jin et al. 2020.
>
> **A:** Great question. Recall that the minimax optimization is equivalent to minimize the envelope function $\Phi(x)$. A second-order stationary point $x$ of $\Phi$ is defined as
> $$
>     \nabla \Phi(x)=\nabla_1 f(x,y^*(x))=\mathbf{0},
> $$
> $$
>     \nabla^2\Phi(x)=\nabla_{11} f(x,y^*(x))-\nabla_{12} f(x,y^*(x))\nabla_{22} f(x,y^*(x))^{-1}\nabla_{21} f(x,y^*(x))\succeq \mathbf{0},
> $$
> which along with the strong concavity condition $\nabla_{22}  f(x,y) \preceq -\mu\mathbf{I}$ implies that $(x,y^*(x))$ satisfies the first-order and second-order necessary conditions defined in Jin et al. 2020. (See their propositions 18,19,20.)
>
> **Q2:** The paper simply applies cubic regularization to the envelope function and thus is not novel.
>
> **A:** Although the idea looks simple, it is a natural and plausible idea from an optimization perspective.
> Despite the simplicity of the idea, this is the first work that develops a convergent algorithm for finding second-order stationary points in nonconvex minimax optimization and provides a comprehensive global/local convergence rate analysis.
>
> **Q3:** We need to discuss more on current first-order methods for the nonconvex-strongly-concave minimax problems. Also, why do the existing results on inexact CR are not directly applicable?
>
> **A:** We appreciate the reviewer's suggestion. In the revision, we have cited and discussed more related works on first-order GDA-type algorithms for nonconvex-strongly-concave problems.
>
> While our analysis can be viewed as an inexact-type analysis of CR, it is different from the existing analysis of inexact CR in several perspectives. First, in the existing literature on inexact CR, the inexactness is often induced by stochastic subsampling when the objective function takes a finite-sum form over a large number of samples. As a comparison, our inexactness is induced by solving the strongly concave problem via gradient ascent. Second, conventional inexact CR usually controls the level of inexactness introduced in every cubic update below a pre-determined constant tolerance level $\epsilon$ in order to guarantee global convergence, but this cannot establish a proper decreasing potential function, which is the key to develop all the convergence analysis of this work.
> As a comparison, our analysis adjust the level of inexactness in every iteration by properly controlling the number of gradient ascent steps. This helps establish the potential function in Proposition 2 that is the key to establish all the global/local convergence results under various types of nonconvex geometries.
>
>
>
> **Q4:** Stepsize $\eta_x$ depends on $\mathcal{O}(\kappa^{-7})$.
>
> **A:** Great question. We found that the dependence of our stepsize on the condition number can be substantially improved, from $\mathcal{O}(\kappa^{-7})$ to $\mathcal{O}(\kappa^{-3})$, by simply optimizing the
> coefficients introduced in several elementary AM-GM inequalities. Specifically, in the revision, we update the proof of the key Proposition 2 and optimize the use of AM-GM inequalities in eqs.(20, 23) to obtain the best dependence of $\eta_x$ on $\kappa$. We have also updated the other results accordingly. Please refer to our revised version for the details.
>
>
> **Q5:** Can cubic-GDA obtain as good results in the local neighborhood of a strict saddle point or a strict local minimum as the original CR method in (Nesterov and Polyak, 2006)?
>
> **A:** Great question. Yes, Cubic-GDA do achieve similar local convergence rate as that of the original CR. For example, it is shown in (Nesterov and Polyak, 2006) that CR achieves a super-linear convergence under the gradient dominant geometry (corresponds to $\theta = 2$ in our Table 1) when sufficiently close to the second-order stationary point. Our Cubic-GDA also enjoys such a fast convergence rate under the same geometry (see our Table 1).
>
> We appreciate the reviewer for pointing out the minor issues. We have addressed them in the revision.

---

### Official Review · Reviewer_N6EU · 2021-11-06

**Correctness:** 4
**Technical Novelty And Significance:** 4
**Empirical Novelty And Significance:** Not applicable
**Recommendation:** 5
**Confidence:** 4

**Main Review:**

To the best of my knowledge, this is one of the very first second-order convergence rate results for nonconvex-strongly concave min-max problems (see another similar paper [1]). Both the results and analysis seem to be good to me. The theoretical contribution is also significant for this area.

I have the following concerns:

In my understanding, ICLR is a conference that emphasizes relevance to practice. The optimization focus in the machine learning community is developing algorithms that are scalable for large-scale problems. This work seems to only focus on giving the convergence rates but not care about per-iteration cost. In fact, both forming the Jacobian matrix in the algorithm and solving the cubic subproblems exactly are very expensive in the machine learning context. So, I am not sure of the practical relevance of this paper. I hope that the authors can provide approximate optimization steps for the primal variable vector and provide the iteration complexity in terms of first-order oracle (i.e., gradient evaluation and Jacobian-vector product).

For the results under Łojasiewicz gradient geometry, I hope that the authors can provide some practical cases that satisfy this assumption in the paper, not only just citing some relevant papers.



[1] Luo, Luo, and Cheng Chen. "Finding Second-Order Stationary Point for Nonconvex-Strongly-Concave Minimax Problem." arXiv preprint arXiv:2110.04814 (2021).







**Summary Of The Paper:**

This paper introduced new algorithms for second-order guarantees of nonconvex-strongly concave problems. First, the paper first gave convergence rate guarantees for the general setting; then it gave linear or superlinear convergence rate results under different orders of  Łojasiewicz gradient geometry.

**Summary Of The Review:**

The authors provide the very first second-order convergence rate results for nonconvex-strongly-concave minimax problems. But I believe that the style of the current version is not very suitable for the machine learning community and the ICLR conference as it does not focus on the large scale setting and does not care about practical performance.

---

> ### Author Response · Authors · 2021-11-18
> **Response to Review**
>
> Thank you very much for reviewing our manuscript and providing valuable feedback. Below is a point-to-point response. We have submitted a revised version with all revisions marked in `red`. Please let us know if further clarifications are needed.
>
> **Q1:** ICLR emphasizes relevance to practice. This work only focuses on convergence rates but not care about per-iteration cost. I hope that the authors can provide approximate optimization steps and iteration complexity in terms of first-order oracle.
>
> **A:** We thank the reviewer for the valuable suggestion. We agree that this work focuses on studying the global/local convergence rates of Cubic-GDA under different types of function geometries, and we think this is a fundamental study on nonconvex optimization that falls into the general interest of ICLR.
>
> Regarding the reviewer's comment on approximate optimization, we want to point out that our analysis framework of Cubic-GDA can allow inexactness in formulating the cubic subproblem, by leveraging the large body of existing development on inexact Cubic Regularization. Specifically, instead of solving the cubic subproblem with the exact gradient $\nabla_1 f(x_t, y_{t+1})$ and Jacobian $G(x_t,y_{t+1})$, it suffices to use their inexact versions $p_t$ and $P_t$ that satisfies the conditions
> $||p_t - \nabla_1 f(x_t, y_{t+1})|| \le O(||x_{t+1} - x_{t}||^2)$, $||P_t - G(x_t,y_{t+1})||\le O(||x_{t+1} - x_{t}||)$. Such type of inexactness conditions are widely used and studied in the existing literature, e.g.,
>
> [1] 'Adaptive cubic regularization methods for unconstrained optimization'
>
> Under these inexact conditions, our proof of the key Proposition 2 remains unchanged, except that the coefficients of the term $||x_{t+1} - x_t||^3$ would be slightly different. We have added a remark on inexactness after Proposition 2 in the revision.
>
> On the other hand, computing a gradient and Jacobian that satisfy the above inexact conditions generally depend on the problem structure and may require a separate comprehensive study. For example, one interesting direction is to consider a finite-sum minimax problem $f(x,y)=\frac{1}{n}\sum_{i=1}^n f_i(x,y)$ with a large sample size $n$, then techniques developed for large-scale cubic regularization such as stochastic subsampling or variance reduction can be applied to construct the inexact gradient and Jacobian.
> We have cited the concurrent work pointed out by the reviewer, which characterizes the oracle complexities under a special type of inexactness that approximates the inverse Jacobian using matrix Chebyshev polynomials.
>
>
>
> **Q2:** Add examples that satisfy Lojasiewicz gradient geometry.
>
> **A:** Thanks for the great suggestion. We have added more concrete examples in the revision.
> For example, consider the class of robust machine learning problems that involve the minimax problem $\min_{\theta}\max_{\xi_i} \frac{1}{n} \sum_{i=1}^n \ell(h_{\theta}(\xi_i), y_i)^2 - \frac{\lambda}{2} ||\xi_i - a_i||^2$. Here $(x_i, y_i)$ is the $i$-th data sample that includes, e.g., an image $x_i$ and its label $y_i$, $\xi_i$ denotes the adversarial data, $h_\theta$ is a model parameterized by $\theta$, and $\ell$ denotes the loss function. Such a problem is nonconvex-stongly-concave when $\lambda$ is large.  In particular, as elaborated in the appendix of the paper 'Proximal Alternating Linearized Minimization for Nonconvex and Nonsmooth Problems', the envelop function $\Phi(x) := \max_{\xi_i} \frac{1}{n} \sum_{i=1}^n \ell(h_{\theta}(\xi_i), y_i)^2 - \frac{\lambda}{2} ||\xi_i - a_i||^2$ satisfies the local Lojasiewicz gradient geometry if it is semi-algebraic, which holds if every sample loss $f(\theta, \xi_i) := \ell(h_{\theta}(\xi_i), y_i)^2 - \frac{\lambda}{2} ||\xi_i - a_i||^2$ is semi-algebraic.

---

### Public Comment · ~Minhui_Huang1 · 2021-11-09
**A potential mistake**

Hi authors and reviewers,

Thanks for this great work!

I found eq (26) seems problematic to me when I was reading the proof. To my understanding, eq (24) only implies

\min ||x_{t+1} - x_t || <= (\frac{ H_0 - \Phi^*}{TL_1}) ^{1/3}.

If this is the case, the inequality (iii) at the bottom of page 17 doesn't hold since we only have for some t the above ineq holds. Could the author explain more on this part? Thanks!

---

### Decision · Program_Chairs · 2022-01-20

**Decision:**

Reject

**Comment:**

The authors provide a cubic regularization approach to non-convex concave minimax problems. The reviewers highlight that the paper in its current form is not ready for publication due to issues such as the gap between the theory and the implementable algorithm.